# Distributed Algorithm for Multi-objective Multi-agent Reinforcement Learning

## Abstract

Multi-objective reinforcement learning (MORL) aims to optimize multiple conflicting objectives for a single agent, where finding Pareto-optimal solutions is NP-hard and existing algorithms are often centralized with high computational complexity, limiting their practical applicability. Multi-objective multi-agent reinforcement learning (MOMARL) extends MORL to multiple agents, which not only increases computational complexity exponentially due to the global state-action space, but also introduces communication challenges, as agents cannot continuously communicate with a central coordinator in large-scale scenarios. This necessitates distributed algorithm, where each agent relies only on the information of its neighbors within a limited range rather than depending on the global scale. To address these challenges, we propose a distributed MOMARL algorithm in which each agent leverages only the state of its $\kappa$-hop neighbors and locally adjusts the weights of multiple objectives through a consensus protocol. We introduce an approximated policy gradient that reduces the dependency on global actions and a linear function approximation that limits the state space to local neighborhoods. Each agent $i$'s computational complexity is thus reduced from $\mathcal{O}(|\boldsymbol{\mathcal{S}}||\boldsymbol{\mathcal{A}}|)$ with global state-action space in centralized algorithms to $\mathcal{O}(|\mathcal{S}_{\mathcal{N}_i^\kappa}||\mathcal{A}_i|)$ with $\kappa$-neighborhood state and local action space. We prove that the algorithm converges to a Pareto-stationary solution at a rate of $\mathcal{O}(1/T)$ and demonstrate in simulations for robot path planning that our approach achieves higher multi-objective values than state-of-the-art method.

## 1 Introduction

As real-world tasks grow increasingly complex, many scenarios naturally involve multiple conflicting objectives, motivating the study of multi-objective reinforcement learning (MORL). For instance, in robotic path planning (Zhang et al., 2016), sa robotic agsent may aim to simultaneously minimize path length, avoid collisions, and maximize information collection.

Different from the rapid development of traditional reinforcement learning (RL) (Grondman et al, 2012; Zhang et al, 2021), research on MORL (Ge et al., 2022; Stamenkovic et al, 2022) remains in its infancy due to the inherent conflicts among multiple objectives. Unlike scalar-reward RL, in MORL the improvement of one objective may degrade others, making standard policy optimization insufficient. A common approach to tackle MORL is to assign fixed weights to objectives and reduce the problem to a single-objective RL (Blondin & Hale, 2020); however, this requires prior knowledge of objective importance and may fail to explore the full Pareto front. To address this limitation, a more rigorous metric is Pareto optimality, where no objective can be improved without degrading others.

However, for non-convex MORL problems, finding exact Pareto-optimal solutions is NP-hard. Consequently, practical algorithms aim for $\epsilon$-Pareto stationary solutions (Sener & Koltun, 2018), which provide a necessary condition for approximate Pareto optimality. On the algorithmic side, for MORL problem with continuous action spaces, (Chen et al., 2021) proposed an actor-critic MORL algorithm based on deterministic policy gradients (Silver et al., 2014) to directly optimize multiple objectives. For MORL with discrete action spaces, (Zhou et al., 2024) introduced a unified multi-objective actor-critic framework applicable to both discounted and average-reward settings, where stochastic policy parameters are updated via a multi-gradient descent approach (Désidéri, 2012), ensuring convergence toward $\epsilon$-Pareto stationary solutions.

The aforementioned methods are all directed towards addressing the MORL problem in a centralized setting or for a single agent. However, practical applications of MORL problems often involve multi-agents. For instance, teams of robots need to decide themselves how to explore distinct regions by simultaneously minimizing energy consumption and travel time. In comparison to the MORL problem with single-agent, the multi-objective multi-agent problem (MOMARL) (Rădulescu, 2020) is more intricate as it encompasses not only potential conflicts among different objectives but also

interactions between the distributed agents with limited communication. An intuitive approach to the MOMARL problem is to consider it as a MORL problem with a single agent, where the state and action are represented by the joint states and joint actions of all agents, respectively. However, as the number of agents increases, the size of their joint state-action space grows exponentially, and in large-scale scenarios, agents cannot communicate with a central coordinator continuously. These characteristic renders the current algorithms used for solving MORL problems with a single agent (Chen et al., 2021; Zhou et al., 2024) unsuitable for large-scale scenarios with multi-agents. Consequently, the MOMARL problem poses new challenges to the design of scalable algorithms and their theoretical analysis.

In this paper, we aims to address the following problem: *How to develop a fully distributed algorithm for the MOMARL problem and ensure its convergence to Pareto-stationary of the multi-objective function?* The contributions of this paper are described as follows.

1. We first propose a novel approximated policy gradient for each agent $i$, which reduces the global action $\boldsymbol{a}$ required by centralized algorithms to the agent's local action $a_i$. Furthermore, to reduce the dimensionality of state information, we employ a linear function approximation to restrict agent $i$'s state to the neighborhood state $s_{\mathcal{N}_i^\kappa}$, encompassing only its $\kappa$-hop neighbors. The per-agent computational complexity is thus reduced from $\mathcal{O}(|\boldsymbol{\mathcal{S}}||\boldsymbol{\mathcal{A}}|)$ in centralized algorithms to $\mathcal{O}(|\mathcal{S}_{\mathcal{N}_i^\kappa}||\mathcal{A}_i|)$.

2. We propose a novel distributed algorithm in which each agent only uses the policy gradient information from its immediate neighbors, while collaboratively adjusting the weights of multiple objectives via a consensus protocol. This design enables the agents to perform cooperative optimization toward a Pareto-stationary solution, without requiring access to the policy gradient information of agents beyond the direct neighbors.

3. We prove that the proposed distributed algorithm, despite relying only on local neighborhood information, achieves convergence to an $\epsilon$-Pareto-stationary solution at a rate of $\mathcal{O}(1/T)$, matching the convergence speed of centralized algorithms. Moreover, we run simulations in a robot path planning environment and show our algorithm converges to greater multi-objective values as compared to the extension of the latest MORL algorithm (Zhou et al., 2024), and performs close to the central optimum with much shorter running time.

For the sake of convenience, some key functions in this paper are presented in Table 1.

Table 1: Symbols and functions.

| Symbols | Annotation |
| --- | --- |
| $Q^m(\boldsymbol{s}, \boldsymbol{a}; \boldsymbol{\theta})$ | Global $Q$-function in the $m$-th objective under joint policy $\boldsymbol{\pi_\theta}$ |
| $Q_i^m(\boldsymbol{s}, \boldsymbol{a}; \boldsymbol{\theta})$ | Local $Q$-function of agent $i$ in the $m$-th objective under joint policy $\boldsymbol{\pi_\theta}$ |
| $Q_{\mathrm{tru},i}^m(s_{\mathcal{N}^\kappa}, a_{\mathcal{N}^\kappa}; \boldsymbol{\theta})$ | Graph-truncated $Q$-function of agent $i$ in the $m$-th objective under joint policy $\boldsymbol{\pi_\theta}$ |
| $\nabla_{\theta_i} J_{\mathrm{tru},i}^m(\boldsymbol{\theta})$ | Graph-truncated policy gradient of agent $i$ in the $m$-th objective under joint policy $\boldsymbol{\pi_\theta}$ |
| $\widehat{Q_i^m}(\boldsymbol{s}, a_i; \boldsymbol{\theta})$ | Action-averaged $Q$-function of agent $i$ in the $m$-th objective under policy $\boldsymbol{\pi_\theta}$ |
| $\nabla_{\theta_i} J_{\mathrm{app},i}^m(\boldsymbol{\theta})$ | Approximated policy gradient of agent $i$ in the $m$-th objective under joint policy $\boldsymbol{\pi_\theta}$ |
| $\hat{Q}_i^m(s_{\mathcal{N}^\kappa}, a_i; w_i^m)$ | Linear approximation function of agent $i$ in the $m$-th objective |

## 2 THE NEW MOMARL PROBLEM FORMULATION AND PRELIMINARIES

### 2.1 MODEL OF THE MOMARL PROBLEM

The MOMARL problem is described as $(\mathcal{N}, \mathcal{M}, \mathcal{G}(\mathcal{N}, \mathcal{E}), \{\mathcal{S}_i\}_{i\in\mathcal{N}}, \{\mathcal{A}_i\}_{i\in\mathcal{N}}, \{\mathcal{P}_i\}_{i\in\mathcal{N}}, \boldsymbol{\rho}, \{r_i^m\}_{i\in\mathcal{N}, m\in\mathcal{M}}, \boldsymbol{\gamma})$, where $\mathcal{N} = \{1, \cdots, N\}$ and $\mathcal{M} = \{1, \cdots, M\}$ represent the agent set and the objective set, respectively. $\mathcal{G} = (\mathcal{N}, \mathcal{E})$ represents the communication network among agents with $\mathcal{E}$ being the set of edges[1]. For integer $\kappa \geq 1$, denote $\mathcal{N}_i^\kappa$ as the $\kappa$-hop neighborhood of agent $i$ and $\mathcal{N}_{-i}^\kappa = \mathcal{N} \setminus \mathcal{N}_i^\kappa$.

**State and action**: $\mathcal{S}_i$ and $\mathcal{A}_i$ represent the local state space and the local action space of agent $i$, respectively. Denote $\boldsymbol{\mathcal{S}} = \prod_{i=1}^N \mathcal{S}_i$ and $\boldsymbol{\mathcal{A}} = \prod_{i=1}^N \mathcal{A}_i$ as the global state space and the global action space, respectively. Denote $\boldsymbol{s} = (s_1, \cdots, s_N) \in \boldsymbol{\mathcal{S}}$ and $\boldsymbol{a} = (a_1, \cdots, a_N) \in \boldsymbol{\mathcal{A}}$ as the global state and the global action of agents, where $s_i \in \mathcal{S}_i$ and $a_i \in \mathcal{A}_i$ represent the local state and local action

---

[1]For the case of time-varying neighbor agents, our algorithm is still applicable if the agent communicates intermittently (or delays communication) with its initial neighbor. In the process of convergence analysis of the algorithm, we just need to introduce an additional error to (22) caused by communication disconnection or delay.

of agent $i \in \mathcal{N}$, respectively. For integer $\kappa \geq 1$, denote $s_{\mathcal{N}_i^\kappa}$ and $a_{\mathcal{N}_i^\kappa}$ as the state and action of agent $i$'s $\kappa$-hop neighbors, respectively. Denote $\mathcal{S}_{\mathcal{N}_i^\kappa} = \prod_{j \in \mathcal{N}_i^\kappa} \mathcal{S}_j$ and $\mathcal{A}_{\mathcal{N}_i^\kappa} = \prod_{j \in \mathcal{N}_i^\kappa} \mathcal{A}_j$ as the state space and the action space of agent $i$'s $\kappa$-hop neighbors, respectively. Moreover, denote $\mathcal{S}_{\mathcal{N}_{-i}^\kappa}$ and $\mathcal{A}_{\mathcal{N}_{-i}^\kappa}$ as the state space and the action space of agents excluding agent $i$'s $\kappa$-hop neighbors, respectively. $a_{\mathcal{N}_{-i}^\kappa} \in \mathcal{S}_{\mathcal{N}_{-i}^\kappa}$

**State transition probability function**: $\mathcal{P}_i(s_i'|s_{\mathcal{N}_i^1}, a_i) : \mathcal{S}_{\mathcal{N}_i^1} \times \mathcal{A}_i \times \mathcal{S}_i \rightarrow [0,1]$ is the state transition probability function of agent $i$, dependent of its 1-hop neighborhood state and its local action. Denote $\boldsymbol{\mathcal{P}}(\boldsymbol{s}'|\boldsymbol{s}, \boldsymbol{a}) = \prod_{i=1}^{N} \mathcal{P}_i(s_i'|s_{\mathcal{N}_i^1}, a_i) : \boldsymbol{\mathcal{S}} \times \boldsymbol{\mathcal{A}} \times \boldsymbol{\mathcal{S}} \rightarrow [0,1]$ as the global state transition probability function. Note that the definition of the state transition probability function $\prod_{i=1}^{N} \mathcal{P}_i(s_i'|s_{\mathcal{N}_i^1}, a_i)$ is common in the literature. For example, it applies to the scenario of traffic signal control problem (Chu et al., 2020), where the traffic flow at each intersection is influenced by the traffic flow at its neighboring intersections and its own signal light.

**Initial state distribution**: $\rho$ is the distribution of the initial state $\boldsymbol{s}_0$.

**Reward function**: $r_i^m(s_i, a_i) : \mathcal{S}_i \times \mathcal{A}_i \rightarrow \mathbb{R}$ is the reward function of agent $i \in \mathcal{N}$ in the objective $m \in \mathcal{M}$. Denote $\boldsymbol{s}_t = (s_{1,t}, \cdots, s_{N,t})$ and $\boldsymbol{a}_t = (a_{1,t}, \cdots, a_{N,t})$ as the global state and the global action at time $t$, respectively. The reward of agent $i \in \mathcal{N}$ in the objective $m \in \mathcal{M}$ at time $t$ can be represented as $r_{i,t}^m = r_i^m(s_{i,t}, a_{i,t})$.

**Discount factor**: $\boldsymbol{\gamma} = (\gamma^1, \cdots, \gamma^M)^\top \in \mathbb{R}^M$ with $\gamma^m \in (0,1)$ being the discount factor in the objective $m \in \mathcal{M}$.

**Softmax policy**: In this paper, we use the parameterized softmax policy $\pi_{\theta_i}(a_i|s_i)$ with parameter $\theta_i \in \mathbb{R}^{|\mathcal{S}_i||\mathcal{A}_i|}$, which is described as

$$\pi_{\theta_i}(a_i|s_i) = \frac{\exp(\theta_{i,s_i,a_i})}{\sum_{a_i'} \exp(\theta_{i,s_i,a_i'})}, \tag{1}$$

where $\theta_{i,s_i,a_i}$ represents the element corresponding to $(s_i, a_i)$ in $\theta_i$. Denote $\boldsymbol{\theta} = (\theta_1^\top, \cdots, \theta_N^\top)^\top \in \mathbb{R}^{\sum_{i=1}^N |\mathcal{S}_i||\mathcal{A}_i|}$ as the joint policy parameter of agents and $\boldsymbol{\pi_\theta}(\boldsymbol{a}|\boldsymbol{s}) = \prod_{i=1}^{N} \pi_{\theta_i}(a_i|s_i)$ be the joint policy of all agents. Note that the softmax policy is used in RL to ensure the exploration of agents (Zhou et al., 2023; Zhang et al., 2022).

In the MOMARL problem, given a joint policy parameter $\boldsymbol{\theta}$, the $m$-th objective of all agents is defined as

$$J^m(\boldsymbol{\theta}) = \mathbb{E}_{\boldsymbol{s} \sim \boldsymbol{\rho}} \Big[ \frac{1}{N} \sum_{t=0}^{\infty} \sum_{i=1}^{N} (\gamma^m)^t r_{i,t}^m | \boldsymbol{s}_0 = \boldsymbol{s}, \boldsymbol{a}_t \sim \boldsymbol{\pi_\theta}(\cdot|\boldsymbol{s}_t) \Big], \tag{2}$$

which is represents the average discounted reward of all agents over all time $t$. The goal of agents in the MOMARL problem is to find a joint policy parameter $\boldsymbol{\theta}$ to maximize the following composite objective, i.e.,

$$\max_{\boldsymbol{\theta}} \boldsymbol{J}(\boldsymbol{\theta}) = [J^1(\boldsymbol{\theta}), \cdots, J^M(\boldsymbol{\theta})]^\top \in \mathbb{R}^M. \tag{3}$$

In order to address the potential conflicts among the $\boldsymbol{J}(\boldsymbol{\theta})$ in (3), the notions of Pareto-optimality and $\epsilon$-Pareto-stationarity are introduced as follows.

**Definition 1** *(Pareto-optimality) A solution $\boldsymbol{\theta}$ dominates solution $\boldsymbol{\theta}'$ if and only if $J^m(\boldsymbol{\theta}) \geq J^m(\boldsymbol{\theta}')$, $\forall m \in \mathcal{M}$ and $\exists m' \in \mathcal{M}$, $J^{m'}(\boldsymbol{\theta}) > J^{m'}(\boldsymbol{\theta}')$. A solution $\boldsymbol{\theta}$ is Pareto-optimal if it is not dominated by any other solution.*

Considering that finding Pareto-optimal solutions for non-convex MOMARL problems is NP-hard, it is generally more practical to seek the $\epsilon$-Pareto-stationary solution instead of the Pareto-optimal solution (Kumar et al., 2019).

**Definition 2** *($\epsilon$-Pareto-stationarity) Define $\nabla_{\boldsymbol{\theta}} \boldsymbol{J}(\boldsymbol{\theta})$ as the gradient of $\boldsymbol{J}(\boldsymbol{\theta})$ respect to $\boldsymbol{\theta}$. A solution $\boldsymbol{\theta}$ is $\epsilon$-Pareto stationary if there exists $\boldsymbol{\lambda} = (\lambda^1, \cdots, \lambda^M)^\top \in \mathbb{R}^M$ such that $\min_{\boldsymbol{\lambda} \in \mathbb{R}^M} \|\nabla_{\boldsymbol{\theta}} \boldsymbol{J}(\boldsymbol{\theta})^\top \boldsymbol{\lambda}\|_2^2 \leq \epsilon$ with $\boldsymbol{\lambda} \geq 0$, $\|\boldsymbol{\lambda}\|_1 = 1$, and $\epsilon > 0$.*

Based on Definitions 1-2, it is obvious that the Pareto-stationarity is a necessary condition for a solution to be Pareto-optimal. Specifically, in the context of convex MOMARL problems, the solutions that are Pareto-stationary also qualify as Pareto-optimal. Given the complexity associated with the MOMARL problem, this paper focuses on developing a distributed scalable algorithm to identify and achieve Pareto-stationarity.

## 2.2 PRELIMINARIES IN THE MOMARL PROBLEM

In the MOMARL problem, for any joint policy parameter $\boldsymbol{\theta}$ and $m \in \mathcal{M}$, the global $Q$-function $Q^m(\boldsymbol{s}, \boldsymbol{a}; \boldsymbol{\theta})$ in $m$-th objective is defined as

$$Q^m(\boldsymbol{s}, \boldsymbol{a}; \boldsymbol{\theta}) = \mathbb{E}_{\boldsymbol{\pi_\theta}} \Big[ \frac{1}{N} \sum_{t=0}^{\infty} \sum_{i=1}^{N} (\gamma^m)^t r_{i,t}^m | \boldsymbol{s}_0 = \boldsymbol{s}, \boldsymbol{a}_0 = \boldsymbol{a} \Big], \tag{4}$$

which represents the value of the stat-action pair $(\boldsymbol{s}, \boldsymbol{a})$ in $m$-th objective under join policy $\boldsymbol{\pi_\theta}$. In the MOMARL problem, given the joint policy parameter $\boldsymbol{\theta}$, define $d_{\boldsymbol{\rho}}^{\boldsymbol{\theta},m}(\boldsymbol{s})$ as the discounted state visitation distribution, which is represented as

$$d_{\boldsymbol{\rho}}^{\boldsymbol{\theta},m}(\boldsymbol{s}) = (1 - \gamma^m) \sum_{t=0}^{\infty} (\gamma^m)^t \mathrm{Pr}^{\boldsymbol{\pi_\theta}}(\boldsymbol{s}_t = \boldsymbol{s} | \boldsymbol{s}_0 \sim \boldsymbol{\rho}), \tag{5}$$

where $\mathrm{Pr}^{\boldsymbol{\pi_\theta}}(\boldsymbol{s}_t = \boldsymbol{s} | \boldsymbol{s}_0 \sim \boldsymbol{\rho})$ represents the probability of $\boldsymbol{s}_t = \boldsymbol{s}$ at time $t$ under the initial state distribution $\boldsymbol{\rho}$ and the joint policy $\boldsymbol{\pi_\theta}$.

Recall that the policy gradient theorem (Sutton et al., 2000) is the foundation of algorithm design in RL. Inspired by the theorem, in our MOMARL problem, we also have the following policy gradient lemma.

**Lemma 1** *In the MOMARL problem, for any joint policy parameter $\boldsymbol{\theta}$, the gradient of $J^m(\boldsymbol{\theta})$ in $m$-the objective with respect to $\boldsymbol{\theta}$ is given by:*

$$\nabla_{\boldsymbol{\theta}} J^m(\boldsymbol{\theta}) = \frac{1}{1 - \gamma^m} \mathbb{E}_{\boldsymbol{s} \sim d_{\boldsymbol{\rho}}^{\boldsymbol{\theta},m}, \boldsymbol{a} \sim \boldsymbol{\pi_\theta}} [\nabla_{\boldsymbol{\theta}} \log \boldsymbol{\pi_\theta}(\boldsymbol{a} | \boldsymbol{s}) Q^m(\boldsymbol{s}, \boldsymbol{a}; \boldsymbol{\theta})], \forall m \in \mathcal{M}. \tag{6}$$

For the policy gradients involved in Definition 2, Lemma 1 shows that the calculation of the policy gradient $\nabla_{\boldsymbol{\theta}} J^m(\boldsymbol{\theta})$ depends on $Q^m(\boldsymbol{s}, \boldsymbol{a}; \boldsymbol{\theta})$, which involves global state-action $(\boldsymbol{s}, \boldsymbol{a})$. Consequently, there are two challenges in applying (6): (i) the computational complexity of handling the global state-action $(\boldsymbol{s}, \boldsymbol{a})$ in a centralized setting is high; (ii) achieving distributed decision making among multi-agents with limited communication.

# 3 DISTRIBUTED SCALABLE ACTOR-CRITIC ALGORITHM FOR MOMARL PROBLEM

Before presenting our proposed method, we first revisit the centralized approach for solving MO-MARL problems. In the centralized setting, the policy update requires the global state-action pair $(\boldsymbol{s}, \boldsymbol{a})$. This immediately leads to an exponential growth of the joint state-action space with the number of agents, i.e., of order $\mathcal{O}(|\boldsymbol{\mathcal{S}}||\boldsymbol{\mathcal{A}}|)$. Such computational complexity makes centralized algorithms prohibitive for large-scale systems. Moreover, centralized training implicitly assumes that agents can constantly communicate with a central controller, which is unrealistic in many real-world scenarios (e.g., swarm robotics, sensor networks). To overcome these limitations, the natural way forward is to make the algorithm scalable, meaning that the per-agent computational cost should remain polynomial in the state-action dimension of its neighbors within a limited range rather than depending on the global scale. The most effective way to achieve scalability is to design a distributed algorithm, where (i) each agent $i$ makes decisions independently using only $(s_{\mathcal{N}_i^\kappa}, a_i)$, where $s_{\mathcal{N}_i^\kappa}$ is the state from its $\kappa$-hop neighbors and (2) exchanges its local Lagrangian multiplier $\lambda_i$ with its directly neighbors for balancing multiple objectives. This shift from centralized to distributed design is the key idea underlying our algorithm: by restricting each agent's decision to local state-action information and coordinating objective trade-offs through a consensus protocol, we preserve scalability while maintaining rigorous convergence guarantees.

To reduce the reliance of algorithm on global information, we design a distributed algorithm, where each agent (i) estimates a local policy gradient based only on its own action, (ii) leverages a linear approximation restricted to $\kappa$-hop neighborhood states, and (iii) updates multi-objective weights through a consensus protocol to cooperatively approach Pareto-stationary solutions.

## 3.1 A NOVEL APPROXIMATED POLICY

In contrast to the global $Q$-function utilized in (6), which relies on global state-action information, we introduce a novel conceptlthe "action-averaged $Q$-function" for each agent $i$. This formulation leverages rewards from agent $i$'s $\kappa$-hop neighbors to effectively reduce the dependence on the full joint action $\boldsymbol{a}$ by focusing on the local action $a_i$, as defined below:

$$\widehat{Q_i^m}(\boldsymbol{s}, a_i; \boldsymbol{\theta}) = \mathbb{E}_{\boldsymbol{\pi_\theta}} \Big[ \frac{1}{N} \sum_{t=0}^{\infty} (\gamma^m)^t \sum_{j \in \mathcal{N}_i^\kappa} r_j^m(s_{j,t}, a_{j,t}) | \boldsymbol{s}_0 = \boldsymbol{s}, a_{i,0} = a_i \Big]. \tag{7}$$

To approximate the true policy gradient $\nabla_{\boldsymbol{\theta}} J^m(\boldsymbol{\theta})$ in (6), we define $\nabla_{\theta_i} J^m_{app}(\boldsymbol{\theta})$ as an approximated policy gradient for agent $i$, derived using the action-averaged $Q$-function presented in (7), as follows:

$$\nabla_{\theta_i} J^m_{app,i}(\boldsymbol{\theta}) = \frac{1}{1-\gamma^m} \mathbb{E}_{\boldsymbol{s} \sim d^{\boldsymbol{\theta},m}_{\boldsymbol{\rho}}, a_i \sim \pi_{\theta_i}} \left[ \widehat{Q^m_i}(\boldsymbol{s}, a_i; \boldsymbol{\theta}) \nabla_{\theta_i} \log \pi_{\theta_i}(a_i|s_i) \right]. \tag{8}$$

Unlike the policy gradient in centralized algorithm that requires global action $\boldsymbol{a}$, (8) only requires the local action $a_i$ of agent $i$. The approximation error between $\nabla_{\theta_i} J^m_{app,i}(\boldsymbol{\theta})$ and original $\nabla_{\theta_i} J^m(\boldsymbol{\theta})$ in (6) can be well bounded for the MOMARL problem in the following theorem.

**Theorem 1** *In the MOMARL problem, given a joint policy $\boldsymbol{\pi_\theta}$, for any agent $i \in \mathcal{N}$ and objective $m \in \mathcal{M}$, it holds that*

$$\|\nabla_{\theta_i} J^m_{app,i}(\boldsymbol{\theta}) - \nabla_{\theta_i} J^m(\boldsymbol{\theta})\|_2 \leq \frac{\sqrt{2}R}{(1-\gamma^m)^2} (\gamma^m)^{\kappa+1}. \tag{9}$$

The proof of Theorem 1 is provided in Appendix A.2. The policy gradient has been approximated so far by constructing $\widehat{Q^m_i}(\boldsymbol{s}, a_i; \boldsymbol{\theta})$ in (7) and $\nabla_{\theta_i} J^m_{app,i}(\boldsymbol{\theta})$ in (8), which reduces the action dimension of each agent $i$ to its local action $a_i$. However, the expression of $\widehat{Q^m_i}(\boldsymbol{s}, a_i; \boldsymbol{\theta})$ still requires the global state. Therefore, in the following, we will focus on reducing the dimensionality of agents' state information.

### 3.2 CRITIC STEP: LINEAR FUNCTION APPROXIMATION

---

**Algorithm 1:** Linear function approximation

---

1 **Require:** The number of samples $K$, the learning-rate $\eta^m_w$ and $\varepsilon > 0$;
2 **Initialization:** Initialize the $\boldsymbol{\varepsilon}$-exploration policy $\boldsymbol{\pi^\varepsilon_\theta} = \Pi^N_{i=1} \pi_{\theta_i}$, where $\pi^\varepsilon_{\theta_i}(a_i|s_i) = (1-\varepsilon)\pi_{\theta_i}(a_i|s_i) + \frac{\varepsilon}{|\mathcal{A}_i|}$ for all $i \in \mathcal{N}$. The initial values of the parameters $w^m_{i,0}$ is set as $w^m_{i,0} = \mathbf{0}_{d_i}$ for all $i \in \{1, 2, \cdots, N\}$;
3 The agents execute the $\boldsymbol{\varepsilon}$-exploration policy $\boldsymbol{\pi^\varepsilon_\theta}$ and each agent $i \in \mathcal{N}$ collects a sequence of samples $\{(s_{i,k}, a_{i,k}, r^m_{i,k})\}_{0 \leq k \leq K}$ in $m$-the objective;
4 **for** $i = 1, 2, \cdots, N$ **do**
5      For each objective $m \in \mathcal{M}$, agent $i \in \mathcal{N}$ collects the state information $\{s_j\}_{j \in \mathcal{N}^\kappa_i}$ of its $\kappa$-hop neighbors and reward $\{r^m_j\}_{j \in \mathcal{N}^\kappa_i}$ from its $\kappa$-hop neighbors to form a sample set $\{s_{\mathcal{N}^\kappa_i,k}, a_{i,k}, r^m_{\mathcal{N}^\kappa_i,k}\}_{0 \leq k \leq K}$;
6      **for** $k = 0, 1, 2, \cdots, K-1$ **do**
7          Each agent $i \in \mathcal{N}$ estimates its local temporal difference error:
         $\delta^m_{i,k} = \phi_i(s_{\mathcal{N}^\kappa_i,k}, a_{i,k})^\top w^m_{i,k} - \frac{1}{N} \sum_{j \in \mathcal{N}^\kappa_i} r^m_{j,k} - \gamma^m \phi_i(s_{\mathcal{N}^\kappa_i,k+1}, a_{i,k+1})^\top w^m_{i,k}$;
8          $w^m_{i,k+1} = w^m_{i,k} - \eta^m_w \delta^m_{i,k} \phi_i(s_{\mathcal{N}^\kappa_i,k+1}, a_{i,k+1})$;
9      **end**
10 **end**
11 **Output:** $\{w^m_{i,K}\}_{i \in \mathcal{N}, m \in \mathcal{M}}$

---

In this subsection, we use the localized stochastic approximation and propose a linear function in (10) to reduce the dimension of the state-action required by agent $i \in \mathcal{N}$ to $(s_{\mathcal{N}^\kappa_i}, a_i)$. Specially, the linear function $\hat{Q}^m_i(s_{\mathcal{N}^\kappa_i}, a_i; w^m_i)$ of agent $i$ to approximate $\widehat{Q^m_i}(\boldsymbol{s}, a_i; \boldsymbol{\theta})$ in (7) is given as

$$\hat{Q}^m_i(s_{\mathcal{N}^\kappa_i}, a_i; w^m_i) = \phi_i(s_{\mathcal{N}^\kappa_i}, a_i)^\top w^m_i, \tag{10}$$

where $\phi_i(s_{\mathcal{N}^\kappa_i}, a_i) : \mathcal{S}_{\mathcal{N}^\kappa_i} \times \mathcal{A}_i \to \mathbb{R}^{d_i}$ is the feature vector mapping and $w^m_i \in \mathbb{R}^{d_i}$ is the parameter of agent $i$ in $m$-th objective. By the definition of $\widehat{Q^m_i}(\boldsymbol{s}, a_i; \boldsymbol{\theta})$ in (7), the parameter with initial value $w^m_{i,0}$ can be updated by sample sequence $\{s_{\mathcal{N}^\kappa_i,k}, a_{i,k}, r^m_{\mathcal{N}^\kappa_i,k}\}_{0 \leq k \leq K}$ as

$$w^m_{i,k+1} = w^m_{i,k} - \eta^m_w \delta^m_{i,k} \phi_i(s_{\mathcal{N}^\kappa_i,k+1}, a_{i,k+1}), \tag{11}$$

where $K$ is the number of sample for linear parameter training and $\delta^m_{i,k}$ is the local temporal difference error, which is discribed as

$$\delta^m_{i,k} = \phi_i(s_{\mathcal{N}^\kappa_i,k}, a_{i,k})^\top w^m_{i,k} - \frac{1}{N} \sum_{j \in \mathcal{N}^\kappa_i} r^m_{j,k} - \gamma^m \phi_i(s_{\mathcal{N}^\kappa_i,k+1}, a_{i,k+1})^\top w^m_{i,k}, \tag{12}$$

and $\eta^m_w$ is the fixed learning rate of parameters $w^m_i$. The detailed description of linear function approximation is given in Algorithm 1.

### 3.3 ACTOR STEP: POLICY PARAMETER UPDATE

To estimate the policy gradient and ensure convergence toward a Pareto-stationary solution, we need to dynamically adjust the weights associated with multiple objectives. In centralized algorithms, this requires access to all agents' policy gradients, which is often infeasible and unscalable (Zhou et al., 2024). To overcome this limitation, we design a distributed algorithm in which agents collaboratively adjust the objective weights through a consensus protocol. This design removes the dependence on global policy gradient while still enabling cooperative optimization toward Pareto-stationary solution.

To estimate the approximated policy gradient $\nabla_{\theta_i} J^m_{app,i}(\boldsymbol{\theta})$ in (8), for joint policy $\boldsymbol{\pi}_{\boldsymbol{\theta}_t}$, our estimate $g^m_{i,t}(B)$ is calculated iteratively based on the sample sequence $\{(s^b_{\mathcal{N}^\kappa_i,h}, a^b_{i,h})\}_{0 \le b \le B-1, 0 \le h \le H-1}$:

$$g^m_{i,t}(b+1) = \frac{b}{b+1} g^m_{i,t}(b) + \frac{1}{b+1} \widehat{\nabla}_{\theta_i} J^{m,b}_{app,i}(\boldsymbol{\theta}_t), \tag{13}$$

where $g^m_{i,t}(0) = \mathbf{0}_{|\mathcal{S}_i||\mathcal{A}_i|}$ and

$$\widehat{\nabla}_{\theta_i} J^{m,b}_{app,i}(\boldsymbol{\theta}_t) = \sum_{h=0}^{H-1} (\gamma^m)^h \phi_i(s^b_{\mathcal{N}^\kappa_i,h}, a^b_{i,h})^\top w^m_i(t) \nabla_{\theta_i} \log \pi_{\theta_{i,t}}(a^b_{i,h}|s^b_{i,h}) \tag{14}$$

with $w^m_i(t)$ being the output of Algorithm 1 in $t$-th iteration of policy parameters. Let $g^m_{i,t} = g^m_{i,t}(B)^\top$ and $\boldsymbol{g}^m_t = \left((g^m_{1,t})^\top, \cdots, (g^m_{N,t})^\top\right)^\top \in \mathbb{R}^{\sum_{i=1}^N |\mathcal{S}_i||\mathcal{A}_i|}$. Following Pareto-stationarity in Definition 1, we denote $\boldsymbol{\lambda}^*_t = (\lambda^{*1}_t, \cdots, \lambda^{*M}_t)^\top \in \mathbb{R}^M$ as solution of the following quadratic programming problem:

$$\min_{\boldsymbol{\lambda}_t = (\lambda^1_t, \cdots, \lambda^M_t)^\top \in \mathbb{R}^M} J^g_t(\boldsymbol{\lambda}_t) = \left\| \sum_{m=1}^M \lambda^m_t \boldsymbol{g}^m_t \right\|^2_2$$
$$\text{s.t. } \boldsymbol{\lambda}_t \ge 0, \|\boldsymbol{\lambda}_t\|_1 = 1. \tag{15}$$

For the network $\mathcal{G}(\mathcal{N}, \mathcal{E})$ among agents, we define its weight matrix as $W^{\mathcal{G}} = [w^{\mathcal{G}}_{ij}]_{N \times N}$, where each element $w^{\mathcal{G}}_{ij}$ represents the weight of the edge from agent $j$ to agent $i$, which is defined as

$$w^{\mathcal{G}}_{ij} = \begin{cases} \frac{1}{1 + \max(|\mathcal{N}_i|, |\mathcal{N}_j|)}, & j \in \mathcal{N}_i, \\ 1 - \sum_{l \in \mathcal{N}_i} w^{\mathcal{G}}_{il}, & j = i, \\ 0, & \text{otherwise.} \end{cases} \tag{16}$$

By using the definition of $W^{\mathcal{G}}$ in (16), we solve the problem (15) in a distributed way, which is presented in the following Algorithm 2.

---

**Algorithm 2:** Distributed computation to solve problem (15) of objective weight adjustment

1 **Initialization:** Each agent $i \in \mathcal{N}$ sets $\boldsymbol{\lambda}_i(0) = \frac{1}{M} \mathbf{1}_M$ and chooses step sizes $\alpha_k = \frac{2}{k+2}$ for all $k \ge 0$;
2 **for** $k = 0, 1, 2, \ldots, K_\lambda - 1$ **do**
3      Each agent $i$ initializes $x_{i,t}(k) = \sum_{m=1}^M \lambda_{i,m}(k) g^m_{i,t}$;
4      **for** $m = 1, 2, \cdots, M$ **do**
5          Each agent $i$ initializes $y^m_{i,t}(k, 0) = \langle x_{i,t}(k), g^m_{i,t} \rangle$;
6          **while** $\exists i \in \mathcal{N}, y^m_{i,t}(l_1) \ne \frac{1}{N} \sum_{i=1}^N y^m_{i,t}(k, 0)$ **do**
7              $y^m_{i,t}(k, l_1 + 1) = \sum_{j \in \mathcal{N}_i} w^{\mathcal{G}}_{ij} y^m_{j,t}(k, l_1)$;
8              $l_1 \leftarrow l_1 + 1$;
9          **end**
10          Each agent $i$ obtains local output $y^m_{i,t}(k)$;
11      **end**
12      Each agent obtains $u^*_{i,t}(k) = \arg\min_m y^m_{i,t}(k)$;
13      Each agent updates $\boldsymbol{\lambda}_i(k+1) = (1 - \alpha_k)\boldsymbol{\lambda}_i(k) + \alpha_k \boldsymbol{e}_{u^*_{i,t}(k)}$, where $\boldsymbol{e}_{u^*_{i,t}(k)}$ is an $M$-dimensional unit vector with the $u^*_{i,t}(k)$-th element being 1, and the other elements being 0;
14 **end**
15 **Output:** $\widehat{\boldsymbol{\lambda}}_t = \boldsymbol{\lambda}_i(K_\lambda)$ for all $i \in \mathcal{N}$;

---

In Algorithm 2 each agent iteratively computes policy gradients, engages in consensus steps to evaluate the quadratic objective (ref. Lines 5-9), and updates its local weight vector via a Frank-Wolfe update rule (ref. Lines 12-13). Through repeated consensus and update steps, all agents asymptot-

ically agree on the optimal weight vector $\widehat{\boldsymbol{\lambda}}_t$, thereby achieving a distributed solution that approximates the centralized Pareto-stationary weighting without requiring global information exchange.

After computing $\widehat{\boldsymbol{\lambda}}_t$ by Algorithm 2, we update the weight $\boldsymbol{\lambda}_t$ as

$$\boldsymbol{\lambda}_t = (1 - \eta_{\boldsymbol{\lambda},t})\boldsymbol{\lambda}_{t-1} + \eta_{\boldsymbol{\lambda},t}\widehat{\boldsymbol{\lambda}}_t, \tag{17}$$

where $\eta_{\boldsymbol{\lambda},t}$ is the learning rate of $\boldsymbol{\lambda}_t$. Denote $\boldsymbol{g}_t = \sum_{m=1}^{M} \lambda_t^m \boldsymbol{g}_t^m$, the update of $\boldsymbol{\theta}_{t+1}$ is presented as

$$\boldsymbol{\theta}_{t+1} = \boldsymbol{\theta}_t + \eta_{\boldsymbol{\theta},t}\boldsymbol{g}_t, \tag{18}$$

where $\eta_{\boldsymbol{\theta},t}$ is the learning rate of policy parameter. In the NMARL problem, the agents can use $\boldsymbol{\theta}_t$ to execute the actions based on (1).

### 3.4 Overall distributed algorithm for MOMARL problem

Based on the distributed designs in the previous three subsections, we propose a distributed MOMARL algorithm, which is given in Algorithm 3.

---

**Algorithm 3:** Distributed algorithm for MOMARL problem

---

1 **Require:** The non-negative integers $T$, $B$, $H$, the learning-rates $\{\eta_{\boldsymbol{\lambda},t}\}_{t \in \{1,\cdots,T\}}$ and $\{\eta_{\boldsymbol{\theta},t}\}_{t \in \{1,\cdots,T\}}$;

2 **Initialization:** Initialize $\boldsymbol{\lambda}_0 = \frac{1}{M}\mathbf{1}_M \in \mathbb{R}^M$, the policy parameter $\theta_{i,1} \in \mathbb{R}^{|\mathcal{S}_i| \times |\mathcal{A}_i|}$ to follow Gaussian distribution for all $i \in \{1, 2, \cdots, N\}$;

3 **for** $t = 1, 2, \cdots, T$ **do**

4     Initial policy gradient estimation $g_{i,t}^m(0) = \mathbf{0}_{|\mathcal{S}_i||\mathcal{A}_i|}$ for all $i \in \mathcal{N}$;

5     **Critic step:** All agents use Algorithm 1 and output the weight vectors $\{w_i^m(t)\}_{i \in \mathcal{N}}$;

6     **Actor step:**

7     **for** $b = 0, 1, 2, \cdots, B - 1$ **do**

8         All agents execute the joint policy $\boldsymbol{\pi}_{\boldsymbol{\theta}_t}$ in $H - 1$ horizon;

9         Each agent $i \in \mathcal{N}$ collects a sequence of samples, which includes the state information $\{s_j\}_{j \in \mathcal{N}_i^\kappa}$ from its $\kappa$-hop neighbors and its local action information $a_i$, i.e., $\{(s_{\mathcal{N}_i^\kappa,h}^b, a_{i,h}^b)\}_{0 \le h \le H-1}$;

10         Each agent $i$ estimates the local policy gradient in $m$-th objective according to (13);

11     **end**

12     All agents calculate $g_{i,t}^m = g_{i,t}^m(B)$ by (13) and achieve $\boldsymbol{g}_t^m = \big((g_{1,t}^m)^\top, \cdots, (g_{N,t}^m)^\top\big)^\top$ for all $m \in [M]$;

13     Compute $\widehat{\boldsymbol{\lambda}}_t$ by Algorithm 2 as the approximationn solution to problem (15);

14     Update the weight $\boldsymbol{\lambda}_t$ according to (17);

15     Update the policy parameter $\boldsymbol{\theta}_{t+1}$ according to (18);

16 **end**

17 **Output:** $\boldsymbol{\pi}_{\boldsymbol{\theta}_{\hat{T}}}$ with $\hat{T}$ chosen uniformly from $\{1, \cdots, T\}$

---

Algorithm 3 incorporates linear function approximation in Algorithm 1 and distributed consensus-based adjustment of multiplier objective weights in Algorithm 2 into a unified framework. The primary advantages are as follows: (1) each agent relies solely on state information from its $\kappa$-hop neighbors, thereby avoiding the exponential expansion of the centralized joint state-action space (ref. Line 9); (2) Each agent in Line 13 only uses the policy gradient estimations of its direct neighbors, which eliminates the requirement for a central coordinator while supporting collaborative multi-objective optimization.

## 4 Pareto-Stationary Convergence of Algorithm 3

Before the convergence analysis of the algorithm, some assumptions are introduced in the following.

**Assumption 1** *In the MOMARL problem, for any joint policy parameter $\boldsymbol{\theta}$ and objective $m \in \mathcal{M}$, $\xi_{\boldsymbol{\rho}}^{\boldsymbol{\theta},m}(\boldsymbol{s}, \boldsymbol{a})$ satisfies that*

$$\inf_{\boldsymbol{\theta}} \min_{(\boldsymbol{s},\boldsymbol{a}) \in \boldsymbol{\mathcal{S}} \times \boldsymbol{\mathcal{A}}} \xi_{\boldsymbol{\rho}}^{\boldsymbol{\theta},m}(\boldsymbol{s}, \boldsymbol{a}) > 0. \tag{19}$$

**Assumption 2** *In the MOMARL problem, for any agent $i \in \mathcal{N}$ and objective $m \in \mathcal{M}$, there exists constant $R > 1$ such that the instantaneous reward $r_{i,t}^m$ at time $t \ge 0$ satisfies $|r_{i,t}^m| \le R$.*

**Assumption 3** *In the MOMARL problem, the network $\mathcal{G}(\mathcal{N}, \mathcal{E})$ among agents is connected graph.*

Assumption 1 ensures that for any joint policy $\boldsymbol{\pi}_{\boldsymbol{\theta}}$, $(\boldsymbol{s}, \boldsymbol{a}) \in \boldsymbol{\mathcal{S}} \times \boldsymbol{\mathcal{A}}$ is visited with a non-zero probability, Assumption 2 provides an upper bound on the reward, and Assumption 3 is provided for designing distributed algorithm. Assumptions 1-2 are standard prerequisite for the convergence

analysis of RL algorithms (e.g. (Zhou et al., 2023; Zhang et al., 2022)) and Assumption 3 is commonly in (Olfati-Saber & Murray, 2004).

Our process to prove the Pareto-stationary convergence of Algorithm 3 is as follows: (i) We start establish the smoothness of objective function (i.e., 2); (ii) We from the definition of Pareto-stationarity in Definition 2 and analyze the error between the true gradient $\nabla_{\theta_i} J^m(\boldsymbol{\theta}_t)$ and the calculated gradient $g_{i,t}^m$ in (13) (i.e., Lemma 3); (iii) We control $\boldsymbol{\lambda}_t$ by setting the step size $\eta_{\boldsymbol{\theta},t}$ to ensure that Algorithm 3 converges to Pareto-stationary solution in Theorem 2.

**Lemma 2** *In the MOMARL problem, define $L_J = \max_{m \in \mathcal{M}} \frac{6N}{(1-\gamma^m)^3}$. For any objective $m \in \mathcal{M}$, the objective $J^m(\boldsymbol{\theta})$ is $L_J$-smooth.*

The detailed proof of Lemma 2 can be found in Appendix A.3. Lemma 2 establishes that each individual objective in the MOMARL problem is $L_J$-smooth with respect to the joint policy parameters. This smoothness property implies that the gradient of $J^m(\boldsymbol{\theta})$ does not change abruptly when the policy parameters are slightly perturbed, which is crucial for analyzing the stability and convergence of gradient-based algorithms. In particular, $L_J$-smoothness allows us to control the error propagation when using approximate gradients or local updates in a distributed setting, and it forms a key technical result for deriving convergence rates toward Pareto-stationary solutions.

Define $\varepsilon_{critic}$ below as the linear approximation error in Algorithm 1:

$$\varepsilon_{critic} = \sup_{m \in \mathcal{M}} \sup_{\boldsymbol{\theta}} \sup_{i \in \mathcal{N}} \mathbb{E}\left[\sup_{\boldsymbol{s},a_i} \left|\hat{Q}_i(s_{\mathcal{N}_i^\kappa}, a_i; w_{i,K}^m) - \widehat{Q_i^m}(\boldsymbol{s}, a_i; \boldsymbol{\theta})\right|^2\right]. \tag{20}$$

Based on $\varepsilon_{critic}$, we further define the gradient approximation error in Section 3.3 as

$$\varepsilon_{actor}^m = \underbrace{\frac{8R^2}{(1-\gamma^m)^4}(\gamma^m)^{2\kappa+2}}_{\text{Truncation error}} + \underbrace{\frac{32}{(1-\gamma^m)^2 B} + \frac{8(\gamma^m)^{2H}}{(1-\gamma^m)^4}}_{\text{Sampling error}} + \underbrace{\frac{8\varepsilon_{critic}}{(1-\gamma^m)^2}}_{\text{Linear approximation error}}. \tag{21}$$

**Lemma 3** *In Algorithm 3, for joint policy parameter $\boldsymbol{\theta}_t$, any agent $i \in \mathcal{N}$, and objective $m \in \mathcal{M}$, we have*

$$\mathbb{E}[\|\nabla_{\theta_i} J^m(\boldsymbol{\theta}_t) - g_{i,t}^m\|_2^2] \leq \varepsilon_{actor}^m.$$

The proof of the Lemma 3 is given in Appendix A.4. Lemma 3 provides an upper bound on the error between the true policy gradient and the estimated gradient $g_{i,t}^m$ used by each agent in Algorithm 3. This result quantifies the approximation error induced by the use of a critic or a finite-horizon estimator in computing policy gradients. Importantly, the bound $\varepsilon_{actor}^m$ captures the combined effects of truncation, sampling, and linear function approximation errors. Based on Lemma 3, the convergence of Algorithm 2 is established in the following theorem.

**Proposition 1** *In Algorithm 2, for any iteration $t$, we define $G_t = (\boldsymbol{g}_t^1, \cdots, \boldsymbol{g}_t^m) \in \mathbb{R}^{M \times \sum_{i=1}^N |\mathcal{S}_i||\mathcal{A}_i|}$ and have the following result:*

*(i) In Line 10 of Algorithm 2, $y_{i,t}^m(k) = \frac{1}{N}\sum_{i=1}^N \langle \sum_{m=1}^M \lambda_{i,m}(k) g_{i,t}^m, g_{i,t}^m \rangle$;*

*(ii) In Problem 15, $J_t^g(\cdot)$ is $L_t^g$-Lipschitz continuous with $L_t^g = 2\sigma_{max}^2(G_t)$, where $\sigma_{max}(G_t)$ is the largest singular value of $G_t$;*

*(iii) $|J_t^g(\widehat{\boldsymbol{\lambda}}_t) - J_t^g(\boldsymbol{\lambda}_t^*)| \leq \frac{4L_t^g}{K_\lambda + 1}$.*

The proof of Proposition 1 is presented in Appendix A.5. Proposition 1 guarantees that, despite the distributed nature of the update and the use of only local gradient information from neighbors, the agents can collectively achieve a close approximation to the globally optimal weight allocation for combining multiple objectives. This property is fundamental for ensuring the cooperative convergence of the distributed algorithm toward a Pareto-stationary solution. Based on Proposition 1, the Pareto-stationary convergence of Algorithm 3 is presented in the following theorem.

**Theorem 2** *In Algorithm 3, let $\eta_{\boldsymbol{\theta},t} = \frac{1}{3L_J}$, and $\eta_{\boldsymbol{\lambda},t} = \frac{1}{(t+1)^2}$. Our policy parameter sequences $\{\boldsymbol{\theta}_t\}_{t=1}^T$ generated by Algorithm 3 satisfies:*

$$\frac{1}{T}\sum_{t=1}^T \mathbb{E}[\|\nabla_{\boldsymbol{\theta}} \boldsymbol{J}(\boldsymbol{\theta}_t)^\top \widehat{\boldsymbol{\lambda}}_t\|_2^2] \leq \frac{216N}{(1-\|\boldsymbol{\gamma}\|_\infty)^3 T}\left(1 + \sum_{t=1}^T \eta_{\boldsymbol{\lambda},t}\right) + 5\max_{m \in \mathcal{M}} \varepsilon_{actor}^m$$

$$+ \frac{8}{K_\lambda + 1}\left(\max_{m \in \mathcal{M}}(\varepsilon_{actor}^m)^2 + \max_{m \in \mathcal{M}} \frac{2R^2}{(1-\gamma^m)^4}\right). \tag{22}$$

The proof of Theorem 2 can be found in Appendix A.6. Theorem 2 shows that Algorithm 3 can converge to an approximate Pareto-stationary solution at a rate of $\mathcal{O}(1/T)$ with some approximate error terms. These errors are not significant, as we can control the upper bound of their upper bounds by setting the feature vector in the linear approximation, sample batch $B$, sample size $H$ for policy gradient approximation, and the $K_\lambda$ for the calculation of objective weights.

## 5 SIMULATION EXPERIMENTS

In this section, we employ a path planning for multiple robotics to travel environment analogous to the one described in (Zhou et al., 2023). While (Zhou et al., 2023) examines a road network comprising 8 nodes, our simulation focuses on a larger network containing 18 nodes. Specifically, the path planning problems of $N$ robots (i.e., agents) on a typical acyclic path network in Fig. 1, where the "blue" nodes in $\{b_1, b_2, \cdots, b_5\}$ represent the set of starting nodes for agents. "purple" node, "orange" node, and "green" node in sthe right-hand-side represent the destination nodes of objective 1, objective 2, and objective 3, respectively. In the path planning problem, the local state space $\mathcal{S}_i$ of agent $i \in \mathcal{N}$ is defined as $\mathcal{S}_i = \{b_1, b_2, b_3, b_4, b_5, c_1, c_2, c_3, c_4, c_5, d_1, d_2, d_3, d_4, d_5, e_1, e_2, e_3\}$, and the local action space of agent $i \in \mathcal{N}$ is defined as $\mathcal{A}_i = \{0, 1, 2, 3\}$ with 3 being the maximum out degree of nodes in path network

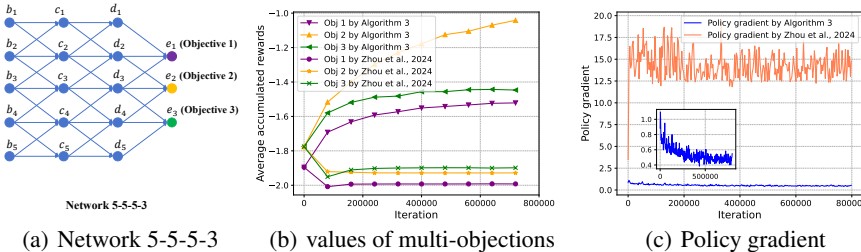

| (a) Network 5-5-5-3 | (b) values of multi-objections | (c) Policy gradient |

Figure 1: (a) Acyclic network. (b,c) The evolution of the objective performance $J(\boldsymbol{\theta}_t)$ and the norm of policy gradient $\|\boldsymbol{g}_t\|_2$ of the policy sequence generated by Algorithm 3, respectively.

In Network 5-5-5-3, consider agent $i$ at node $b_2$ for illustration, the action "0" means that the agent $i$ remains stationary at the current node for one time step, while "1", "2", and "3" indicate that the agent $i$ follows the edge $(b_2, c_1)$, edge $(b_2, c_2)$, and edge $(b_2, c_3)$, respectively. It should be noted that the agent $i$ remains at the current node even when the action selected by it exceeds the out degree of the current node. For example, if agent $j \in \mathcal{N}$ selects the action "3" at node $b_1$, then it will remain stationary at $b_1$ for one time step.

The reward settings of each agent include: (i) the time run cost $-0.5$ at each step; (ii) the collision penalty when the agents share a path to move; (iii) the additional rewards for achieving different objectives. Specifically, when an agent reaches objective 1, objective 2, and objective 3, it receives the additional rewards of 0.5, 1.5, and 1, respectively. In this path planning problem, the agents want to efficiently reach the destinations while avoiding collision. The objective of agents is to find a joint policy parameter $\boldsymbol{\theta}$ to maximizes (3).

Our robot path planning problem includes 10 agents, whose initial positions are set to $\{b_1, b_2, b_3, b_4, b_5, b_1, b_2, b_3, b_4, b_5\}$. In this simulation, both the proposed Algorithm 3 and the latest MORL centralized algorithm from (Zhou et al., 2024) are tested under different random seeds. The discounted average cumulative reward $\{J^m(\boldsymbol{\theta}_t)\}_{m \in \{1,2,3\}}$ of the policy sequence generated by Algorithm 3 and the centralized algorithm are depicted in Fig. 1(b), where Algorithm 3 outperforms the centralized algorithm on performance of each objective. The main reason is that the centralized algorithm applies linear approximation to the global state-action space, causing large errors, whereas our distributed algorithm only approximates local $Q$-functions, leading to higher accuracy.

Moreover, the norm of policy gradients (i.e., $\|\boldsymbol{g}_t\|_2$) generated by Algorithm 3 and the centralized algorithm are showed in Fig. 1(c). The norm of the policy gradient in Algorithm 3 exhibits a fast convergence trend towards to 0. However, the policy gradient in the centralized algorithm does not converge but fluctuates over many iterations from 0 due to the excessively large state-action dimension, resulting in a substantial approximation error in the linear approximation.

## 6 CONCLUSIONS

We propose a distributed algorithm for MOMARL and prove its convergence to a close-to-Pareto-stationary point. Each agent only requires state-action information $(s_{\mathcal{N}_i^\kappa}, a_i)$, ensuring scalability. This framework itself is a significant contribution and may inspire other scalable RL methods in networked systems

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
