_{i,k}^m = \phi_i(s_{\mathcal{N}_i^\kappa,k}, a_{i,k})^\top w_{i,k}^m - \frac{1}{N} \sum_{j \in \mathcal{N}_i^\kappa} r_{j,k}^m - \gamma^m \phi_i(s_{\mathcal{N}_i^\kappa,k+1}, a_{i,k+1})^\top w_{i,k}^m$;
8          $w_{i,k+1}^m = w_{i,k}^m - \eta_w^m \delta_{i,k}^m \phi_i(s_{\mathcal{N}_i^\kappa,k+1}, a_{i,k+1})$;
9      **end**
10 **end**
11 **Output:** $\{w_{i,K}^m\}_{i \in \mathcal{N}, m \in \mathcal{M}}$

---

In this subsection, we use the localized stochastic approximation and propose a linear function in (10) to reduce the dimension of the state-action required by agent $i \in \mathcal{N}$ to $(s_{\mathcal{N}_i^\kappa}, a_i)$. Specially, the linear function $\hat{Q}_i^m(s_{\mathcal{N}_i^\kappa}, a_i; w_i^m)$ of agent $i$ to approximate $\widehat{Q_i^m}(\boldsymbol{s}, a_i; \boldsymbol{\theta})$ in (7) is given as

$$\hat{Q}_i^m(s_{\mathcal{N}_i^\kappa}, a_i; w_i^m) = \phi_i(s_{\mathcal{N}_i^\kappa}, a_i)^\top w_i^m, \tag{10}$$

where $\phi_i(s_{\mathcal{N}_i^\kappa}, a_i) : \mathcal{S}_{\mathcal{N}_i^\kappa} \times \mathcal{A}_i \to \mathbb{R}^{d_i}$ is the feature vector mapping and $w_i^m \in \mathbb{R}^{d_i}$ is the parameter of agent $i$ in $m$-th objective. By the definition of $\widehat{Q_i^m}(\boldsymbol{s}, a_i; \boldsymbol{\theta})$ in (7), the parameter with initial value $w_{i,0}^m$ can be updated by sample sequence $\{s_{\mathcal{N}_i^\kappa,k}, a_{i,k}, r_{\mathcal{N}_i^\kappa,k}^m\}_{0 \leq k \leq K}$ as

$$w_{i,k+1}^m = w_{i,k}^m - \eta_w^m \delta_{i,k}^m \phi_i(s_{\mathcal{N}_i^\kappa,k+1}, a_{i,k+1}), \tag{11}$$

where $K$ is the number of sample for linear parameter training and $\delta_{i,k}^m$ is the local temporal difference error, which is discribed as

$$\delta_{i,k}^m = \phi_i(s_{\mathcal{N}_i^\kappa,k}, a_{i,k})^\top w_{i,k}^m - \frac{1}{N} \sum_{j \in \mathcal{N}_i^\kappa} r_{j,k}^m - \gamma^m \phi_i(s_{\mathcal{N}_i^\kappa,k+1}, a_{i,k+1})^\top w_{i,k}^m, \tag{12}$$

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

# A  APPENDIX

## A.1  PRELIMINARY DEFINITIONS FOR THEOREM 1

In this subsection, we introduce the formal definition of the exponential decay property in the MO-MARL problem.

Different from the definition of the global $Q$-function in (4), for each agent $i \in \mathcal{N}$, its local $Q$-function $Q_i^m(\boldsymbol{s}, \boldsymbol{a}; \boldsymbol{\theta})$ in $m$-th objective is defined as

$$Q_i^m(\boldsymbol{s}, \boldsymbol{a}; \boldsymbol{\theta}) = \mathbb{E}_{\boldsymbol{\pi_\theta}} \Big[ \sum_{t=0}^{\infty} (\gamma^m)^t r_{i,t}^m | \boldsymbol{s}_0 = \boldsymbol{s}, \boldsymbol{a}_0 = \boldsymbol{a} \Big]. \tag{23}$$

Based on the definitions of the global $Q$-function (4) and the local $Q$-function (23), we have

$$Q^m(\boldsymbol{s}, \boldsymbol{a}; \boldsymbol{\theta}) = \frac{1}{N} \sum_{i=1}^{N} Q_i^m(\boldsymbol{s}, \boldsymbol{a}; \boldsymbol{\theta}), \tag{24}$$

which shows the global $Q$-function can be decomposed into the sum of the local $Q$-functions of all agents.

**Definition 3** *The MOMARL satisfies the $(\boldsymbol{\vartheta}, \boldsymbol{\varrho})$-exponential decay property with $\boldsymbol{\vartheta} = (\vartheta^1, \cdots, \vartheta^M)^\top \in \mathbb{R}^M, \boldsymbol{\varrho} = (\varrho^1, \cdots, \varrho^M)^\top \in \mathbb{R}^M$, if for any joint policy $\boldsymbol{\pi_\theta}$, agent $i \in \mathcal{N}$, objective $m \in \mathcal{M}$, $s_{\mathcal{N}_i^\kappa} \in \mathcal{S}_{\mathcal{N}_i^\kappa}$, $a_{\mathcal{N}_i^\kappa} \in \mathcal{A}_{\mathcal{N}_i^\kappa}$, $s_{\mathcal{N}_{-i}^\kappa}, s'_{\mathcal{N}_{-i}^\kappa} \in \mathcal{S}_{\mathcal{N}_{-i}^\kappa}$, and $a_{\mathcal{N}_{-i}^\kappa}, a'_{\mathcal{N}_{-i}^\kappa} \in \mathcal{A}_{\mathcal{N}_{-i}^\kappa}$, $Q_i^m(\boldsymbol{s}, \boldsymbol{a}; \boldsymbol{\theta})$ satisfies*

$$\Big| Q_i^m(s_{\mathcal{N}_i^\kappa}, s_{\mathcal{N}_{-i}^\kappa}, a_{\mathcal{N}_i^\kappa}, a_{\mathcal{N}_{-i}^\kappa}; \boldsymbol{\theta}) - Q_i^m(s_{\mathcal{N}_i^\kappa}, s'_{\mathcal{N}_{-i}^\kappa}, a_{\mathcal{N}_i^\kappa}, a'_{\mathcal{N}_{-i}^\kappa}; \boldsymbol{\theta}) \Big| \le \vartheta^m (\varrho^m)^{\kappa+1}. \tag{25}$$

The exponential decay property of the MOMARL problem indicates that the dependence of agent $i$'s local $Q$-function $Q_i^m(\boldsymbol{s}, \boldsymbol{a}; \boldsymbol{\theta})$ on other agents shrinks rapidly as the distance between them increases. By Assumption 2, we can directly obtain the following lemma.

**Lemma 4** *The MOMARL problem satisfies $\big( (\frac{R}{1-\gamma^1}, \cdots, \frac{R}{1-\gamma^M})^\top, \boldsymbol{\gamma} \big)$-exponential decay property.*

**Proof.** For any objective $m \in \mathcal{M}$ and agent $i \in \mathcal{N}$, by using Lemma 3 in (Qu et al., 2020a), we have that

$$\Big| Q_i^m(s_{\mathcal{N}_i^\kappa}, s_{\mathcal{N}_{-i}^\kappa}, a_{\mathcal{N}_i^\kappa}, a_{\mathcal{N}_{-i}^\kappa}; \boldsymbol{\theta}) - Q_i^m(s_{\mathcal{N}_i^\kappa}, s'_{\mathcal{N}_{-i}^\kappa}, a_{\mathcal{N}_i^\kappa}, a'_{\mathcal{N}_{-i}^\kappa}; \boldsymbol{\theta}) \Big| \le \frac{R}{1-\gamma^m} (\gamma^m)^{\kappa+1},$$

which can further deduce that the MOMARL problem satisfies the $\big( (\frac{R}{1-\gamma^1}, \cdots, \frac{R}{1-\gamma^M})^\top, \boldsymbol{\gamma} \big)$-exponential decay property. $\qquad \square$

Lemma 4 provides a possibility for agents to approximate $Q_i^m(\boldsymbol{s}, \boldsymbol{a}; \boldsymbol{\theta})$ by only using its $\kappa$-hop neighbors' information. Inspired by exponential decay property in Lemma 4, we design a proper class of graph-truncated $Q$-functions:

$$Q_{tru,i}^m(s_{\mathcal{N}_i^\kappa}, a_{\mathcal{N}_i^\kappa}; \boldsymbol{\theta}) = \sum_{s_{\mathcal{N}_{-i}^\kappa}, a_{\mathcal{N}_{-i}^\kappa}} \xi_{\boldsymbol{\rho}}^{\boldsymbol{\theta}, m}(s_{\mathcal{N}_{-i}^\kappa}, a_{\mathcal{N}_{-i}^\kappa} | s_{\mathcal{N}_i^\kappa}, a_{\mathcal{N}_i^\kappa})$$
$$Q_i^m(s_{\mathcal{N}_i^\kappa}, s_{\mathcal{N}_{-i}^\kappa}, a_{\mathcal{N}_i^\kappa}, a_{\mathcal{N}_{-i}^\kappa}; \boldsymbol{\theta}), \tag{26}$$

where $\xi_{\boldsymbol{\rho}}^{\boldsymbol{\theta}, m}(s_{\mathcal{N}_{-i}^\kappa}, a_{\mathcal{N}_{-i}^\kappa} | s_{\mathcal{N}_i^\kappa}, a_{\mathcal{N}_i^\kappa})$ is the weight coefficient and satisfies

$$\xi_{\boldsymbol{\rho}}^{\boldsymbol{\theta}, m}(s_{\mathcal{N}_{-i}^\kappa}, a_{\mathcal{N}_{-i}^\kappa} | s_{\mathcal{N}_i^\kappa}, a_{\mathcal{N}_i^\kappa})$$
$$= \frac{\xi_{\boldsymbol{\rho}}^{\boldsymbol{\theta}, m}(s_{\mathcal{N}_i^\kappa}, s_{\mathcal{N}_{-i}^\kappa}, a_{\mathcal{N}_i^\kappa}, a_{\mathcal{N}_{-i}^\kappa})}{\sum_{s'_{\mathcal{N}_{-i}^\kappa}, a'_{\mathcal{N}_{-i}^\kappa}} \xi_{\boldsymbol{\rho}}^{\boldsymbol{\theta}, m}(s_{\mathcal{N}_i^\kappa}, s'_{\mathcal{N}_{-i}^\kappa}, a_{\mathcal{N}_i^\kappa}, a'_{\mathcal{N}_{-i}^\kappa})}. \tag{27}$$

Using (26), we define the graph-truncated policy gradient $\nabla_{\theta_i} J_{tru,i}^m(\boldsymbol{\theta})$ as

$$\nabla_{\theta_i} J_{tru,i}^m(\boldsymbol{\theta}) = \mathbb{E}_{\boldsymbol{s} \sim d_{\boldsymbol{\rho}}^{\boldsymbol{\theta}, m}, \boldsymbol{a} \sim \boldsymbol{\pi_\theta}} \Big[ \sum_{j \in \mathcal{N}_i^\kappa} Q_{tru,j}^m(s_{\mathcal{N}_j^\kappa}, a_{\mathcal{N}_j^\kappa}; \boldsymbol{\theta})$$
$$\nabla_{\theta_i} \log \pi_{\theta_i}(a_i | s_i) \Big] \frac{1}{(1-\gamma)N}. \tag{28}$$

The graph-truncated policy gradient approximation error is presented in the following.

**Lemma 5** *In the MOMARL problem, for any agent $i \in \mathcal{N}$ and objective $m \in \mathcal{M}$, we have*

$$\left\| \nabla_{\theta_i} J_{tru,i}^m(\boldsymbol{\theta}) - \nabla_{\theta_i} J^m(\boldsymbol{\theta}) \right\|_2 \leq \frac{\sqrt{2}R}{(1-\gamma^m)^2}(\gamma^m)^{\kappa+1}. \tag{29}$$

The proof of Lemma 5 is similar to Lemma 4 in (Qu et al., 2020a), therefore omitted here. Lemma 5 shows that the graph-truncated $Q$-functions $\{Q_{tru,j}^m(s_{\mathcal{N}_j^\kappa}, a_{\mathcal{N}_j^\kappa}; \boldsymbol{\theta})\}_{j \in \mathcal{N}_i^\kappa}$ can effectively approximate the policy gradient $\nabla_{\theta_i} J^m(\boldsymbol{\theta})$ through the state-action $(s_{\mathcal{N}_i^\kappa}, a_{\mathcal{N}_i^\kappa})$.

Unlike the graph-truncated policy gradient $\nabla_{\theta_i} J_{tru,i}^m(\boldsymbol{\theta})$ in (28) that requires $a_{\mathcal{N}_i^\kappa}$, (8) only requires the local action $a_i$. Next, we establish the equivalence between graph-truncated policy gradient $\nabla_{\theta_i} J_{tru,i}^m(\boldsymbol{\theta})$ and approximated policy gradient $\nabla_{\theta_i} J_{app}^m(\boldsymbol{\theta})$ in the following proposition.

**Proposition 2** *In the MOMARL problem, given a joint policy $\boldsymbol{\pi_\theta}$, for any agent $i \in \mathcal{N}$ and objective $m \in \mathcal{M}$, it holds*

$$\nabla_{\theta_i} J_{tru,i}^m(\boldsymbol{\theta}) = \nabla_{\theta_i} J_{app,i}^m(\boldsymbol{\theta}). \tag{30}$$

**Proof.** By the definition of $Q_{tru,i}^m(s_{\mathcal{N}_i^\kappa}, a_{\mathcal{N}_i^\kappa}; \boldsymbol{\theta})$ in (26), we have

$$\mathbb{E}_{\boldsymbol{s} \sim d_\rho^{\boldsymbol{\theta},m}, \boldsymbol{a} \sim \boldsymbol{\pi_\theta}} \left[ \frac{1}{N} \sum_{j \in \mathcal{N}_i^\kappa} Q_{tru,j}^m(s_{\mathcal{N}_j^\kappa}, a_{\mathcal{N}_j^\kappa}; \boldsymbol{\theta}) \nabla_{\theta_i} \log \pi_{\theta_i}(a_i | s_i) \right]$$

$$= \mathbb{E}_{\boldsymbol{s} \sim d_\rho^{\boldsymbol{\theta},m}, \boldsymbol{a} \sim \boldsymbol{\pi_\theta}} \left[ \frac{1}{N} \sum_{j \in \mathcal{N}_i^\kappa} \sum_{\tilde{s}_{-\mathcal{N}_j^\kappa}, \tilde{a}_{-\mathcal{N}_j^\kappa}} \xi_\rho^{\boldsymbol{\theta},m}(\tilde{s}_{-\mathcal{N}_j^\kappa}, \tilde{a}_{-\mathcal{N}_j^\kappa} | s_{\mathcal{N}_j^\kappa}, a_i, a_{\mathcal{U}_{j,-i}^\kappa}) Q_j^m(s_{\mathcal{N}_j^\kappa}, \tilde{s}_{-\mathcal{N}_j^\kappa}, a_i, a_{\mathcal{U}_{j,-i}^\kappa}, \tilde{a}_{-\mathcal{N}_j^\kappa}; \boldsymbol{\theta}) \right.$$

$$\left. \nabla_{\theta_i} \log \pi_{\theta_i}(a_i | s_i) \right]$$

$$= \mathbb{E}_{\boldsymbol{s} \sim d_\rho^{\boldsymbol{\theta},m}, \boldsymbol{a} \sim \boldsymbol{\pi_\theta}} \left[ \frac{1}{N} \sum_{j \in \mathcal{N}_i^\kappa} Q_j^m(s_{\mathcal{N}_j^\kappa}, s_{-\mathcal{N}_j^\kappa}, a_i, a_{\mathcal{U}_{j,-i}^\kappa}, a_{-\mathcal{N}_j^\kappa}; \boldsymbol{\theta}) \nabla_{\theta_i} \log \pi_{\theta_i}(a_i | s_i) \right] \tag{31}$$

$$= \mathbb{E}_{\boldsymbol{s} \sim d_\rho^{\boldsymbol{\theta},m}, a_i \sim \pi_{\theta_i}} \left[ \frac{1}{N} \mathbb{E}_{\boldsymbol{\pi_\theta}} \left[ \sum_{t=0}^\infty (\gamma^m)^t \sum_{j \in \mathcal{N}_i^\kappa} r_j^m(s_{j,t}, a_{j,t}) | \boldsymbol{s}_0 = \boldsymbol{s}, a_{i,0} = a_i \right] \nabla_{\theta_i} \log \pi_{\theta_i}(a_i | s_i) \right]$$

$$\tag{32}$$

$$= \mathbb{E}_{\boldsymbol{s} \sim d_\rho^{\boldsymbol{\theta},m}, a_i \sim \pi_{\theta_i}} \left[ \widehat{Q_i^m}(\boldsymbol{s}, a_i; \boldsymbol{\theta}) \nabla_{\theta_i} \log \pi_{\theta_i}(a_i | s_i) \right], \tag{33}$$

where the second equality (31) is obtained from the definition of $\xi_\rho^{\boldsymbol{\theta},m}(s_{\mathcal{N}_{-i}^\kappa}, a_{\mathcal{N}_{-i}^\kappa} | s_{\mathcal{N}_i^\kappa}, a_{\mathcal{N}_i^\kappa})$ in (27), the third equality (32) comes from the definition of the local $Q$-function in (23), and the last equality (33) can be achieved by the definition of $\widehat{Q_i^m}(\boldsymbol{s}, a_i; \boldsymbol{\theta})$ in (7). $\qquad\square$

Proposition 2 provides an equivalence between $Q_{tru,i}^m(s_{\mathcal{N}_i^\kappa}, a_{\mathcal{N}_j^\kappa}; \boldsymbol{\theta})$ and $\widehat{Q_i^m}(\boldsymbol{s}, a_i; \boldsymbol{\theta})$ in policy gradient approximation.

### A.2 THE PROOF OF THEOREM 1

**Proof.** By the definition of $\nabla_{\theta_i} J_{app,i}^m(\boldsymbol{\theta})$ in (8), we have

$$\|\nabla_{\theta_i} J_{app,i}^m(\boldsymbol{\theta}) - \nabla_{\theta_i} J^m(\boldsymbol{\theta})\|_2 = \|\nabla_{\theta_i} J_{app,i}^m(\boldsymbol{\theta}) - \nabla_{\theta_i} J_{tru,i}^m(\boldsymbol{\theta}) + \nabla_{\theta_i} J_{tru,i}^m(\boldsymbol{\theta}) - \nabla_{\theta_i} J^m(\boldsymbol{\theta})\|_2$$

$$= \|\nabla_{\theta_i} J_{tru,i}^m(\boldsymbol{\theta}) - \nabla_{\theta_i} J^m(\boldsymbol{\theta})\|_2 \tag{34}$$

$$\leq \frac{\sqrt{2}R}{(1-\gamma^m)^2}(\gamma^m)^{\kappa+1}, \tag{35}$$

where the second equality comes from Proposition 2 and last inequality achieved by Lemma 5. Hence, the proof is completed. $\qquad\square$

### A.3 PROOF OF LEMMA 2

In the MOMARL problem, for any $i \in \mathcal{N}$, denote $s_{-i} = \boldsymbol{s} \setminus s_i$ as the state of agents other than agent $i$ and $a_{-i} = \boldsymbol{a} \setminus a_i$ as the action of agents other than agent $i$. For any joint policy $\boldsymbol{\pi_\theta}$, denote $d_{\rho,i}^{\boldsymbol{\theta},m}(s_i)$ and $d_{\rho,-i}^{\boldsymbol{\theta},m}(s_{-i})$ as the discounted state visitation distribution of $s_i$ and $s_{-i}$ in $m$-objective, respectively. For each agent $i \in \mathcal{N}$, define its local value function $V_i^m(\boldsymbol{s}; \boldsymbol{\theta})$ in $m$-th objective as

$$V_i^m(\boldsymbol{s}; \boldsymbol{\theta}) = \sum_{\boldsymbol{a}} \boldsymbol{\pi_\theta}(\boldsymbol{a} | \boldsymbol{s}) Q_i^m(\boldsymbol{s}, \boldsymbol{a}; \boldsymbol{\theta}). \tag{36}$$

Define the averaged value function, the averaged $Q$-function, and the averaged advantage function of agent $i \in \mathcal{N}$ in the objective $m \in \mathcal{M}$ as

$$\overline{V_i^m}(s_i; \boldsymbol{\theta}) = \frac{1}{N} \sum_{s'_{-i}} d_{\boldsymbol{\rho}, -i}^{\boldsymbol{\theta}, m}(s'_{-i}) \sum_{j \in \mathcal{N}} V_j^m(s_i, s'_{-i}; \boldsymbol{\theta}), \tag{37}$$

$$\overline{Q_i^m}(s_i, a_i; \boldsymbol{\theta}) = \frac{1}{N} \sum_{s'_{-i}, a'_{-i}} d_{\boldsymbol{\rho}, -i}^{\boldsymbol{\theta}, m}(s'_{-i}) \pi_{\theta_{-i}}(a'_{-i}|s'_{-i})$$

$$\sum_{j \in \mathcal{N}} Q_j^m(s_i, s'_{-i}, a_i, a'_{-i}; \boldsymbol{\theta}), \tag{38}$$

$$\overline{A_i^m}(s_i, a_i; \boldsymbol{\theta}) = \overline{Q_i^m}(s_i, a_i; \boldsymbol{\theta}) - \overline{V_i^m}(s_i; \boldsymbol{\theta}). \tag{39}$$

**Lemma 6** *(Softmax policy gradient) In the MOMARL problem, for any joint policy $\boldsymbol{\pi_\theta}$, agent $i \in \mathcal{N}$, and objective $m \in \mathcal{M}$, the gradient of $J^m(\boldsymbol{\theta})$ with respect to $\theta_{i, s_i, a_i}$ is represented as*

$$\frac{\partial J^m(\boldsymbol{\theta})}{\partial \theta_{i, s_i, a_i}} = \frac{1}{1 - \gamma^m} d_{\boldsymbol{\rho}, i}^{\boldsymbol{\theta}, m}(s_i) \pi_{\theta_i}(a_i|s_i) \overline{A_i^m}(s_i, a_i; \boldsymbol{\theta}). \tag{40}$$

**Proof.** According to the policy gradient lemma 1 and (24), we have

$$\frac{\partial J^m(\boldsymbol{\theta})}{\partial \theta_{i, s_i, a_i}}$$

$$= \frac{1}{1 - \gamma^m} \sum_{s', a'} d_{\boldsymbol{\rho}}^{\boldsymbol{\theta}}(s') \boldsymbol{\pi_\theta}(a'|s') \frac{\partial \log \boldsymbol{\pi_\theta}(a'|s')}{\partial \theta_{i, s_i, a_i}}$$

$$\left( \frac{1}{N} \sum_{j=1}^N Q_j^m(s', a'; \boldsymbol{\theta}) \right)$$

$$= \frac{1}{1 - \gamma^m} \sum_{s'_i, a'_i} d_{\boldsymbol{\rho}, i}^{\boldsymbol{\theta}, m}(s'_i) \pi_{\theta_i}(a'_i|s'_i)$$

$$\sum_{s'_{-i}, a'_{-i}} d_{\boldsymbol{\rho}, -i}^{\boldsymbol{\theta}, m}(s'_{-i}) \pi_{\theta_{-i}}(a'_{-i}|s'_{-i}) \Big( \mathbf{1}\{s'_i = s_i, a'_i = a_i\}$$

$$- \mathbf{1}\{s'_i = s_i\} \pi_{\theta_i}(a_i|s_i) \Big) \frac{1}{N} \Big( \sum_{j \in \mathcal{N}} Q_j^m(s_i, s'_{-i}, a_i, a'_{-i}; \boldsymbol{\theta}) \Big) \tag{41}$$

$$= \frac{1}{1 - \gamma^m} d_{\boldsymbol{\rho}, i}^{\boldsymbol{\theta}, m}(s_i) \pi_{\theta_i}(a_i|s_i) \overline{Q_i^m}(s_i, a_i; \boldsymbol{\theta})$$

$$- \frac{1}{1 - \gamma^m} d_{\boldsymbol{\rho}, i}^{\boldsymbol{\theta}, m}(s_i) \pi_{\theta_i}(a_i|s_i) \sum_{a_i} \pi_{\theta_i}(a_i|s_i) \overline{Q_i^m}(s_i, a_i; \boldsymbol{\theta})$$

$$= \frac{1}{1 - \gamma^m} d_{\boldsymbol{\rho}, i}^{\boldsymbol{\theta}, m}(s_i) \pi_{\theta_i}(a_i|s_i) \big( \overline{Q_i^m}(s_i, a_i; \boldsymbol{\theta}) - \overline{V_i^m}(s_i; \boldsymbol{\theta}) \big)$$

$$= \frac{1}{1 - \gamma^m} d_{\boldsymbol{\rho}, i}^{\boldsymbol{\theta}, m}(s_i) \pi_{\theta_i}(a_i|s_i) \overline{A_i^m}(s_i, a_i; \boldsymbol{\theta}), \tag{42}$$

where the second equality (41) comes from the fact that

$$\frac{\partial \log \boldsymbol{\pi_\theta}(a'|s')}{\partial \theta_{i, s_i, a_i}} = \frac{\partial \log \pi_{\theta_i}(a'_i|s'_i)}{\partial \theta_{i, s_i, a_i}}$$

$$= \mathbf{1}\{s'_i = s_i, a'_i = a_i\} - \mathbf{1}\{s'_i = s_i\} \pi_{\theta_i}(a_i|s_i), \tag{43}$$

and the last equality (42) can be obtained from the definition of the averaged advantage function $\overline{A_i^m}(s_i, a_i; \boldsymbol{\theta})$. $\square$ Based on Lemma 6, the proof of Lemma 2 are represented as follows. Consider that for any different policies $\boldsymbol{\pi_\theta}$ and $\boldsymbol{\pi_{\theta'}}$, we have

$$\|\nabla_{\boldsymbol{\theta}} J^m(\boldsymbol{\theta'}) - \nabla_{\boldsymbol{\theta}} J^m(\boldsymbol{\theta})\|_2^2$$

$$\leq \sum_{i=1}^{N} \|\nabla_{\theta_i} J^m(\boldsymbol{\theta}') - \nabla_{\theta_i} J^m(\boldsymbol{\theta})\|_2^2$$

$$\leq \sum_{i=1}^{N} \|\nabla_{\theta_i} J^m(\boldsymbol{\theta}') - \nabla_{\theta_i} J^m(\boldsymbol{\theta})\|_1^2. \tag{44}$$

By Lemmas 6, we have

$$\|\nabla_{\theta_i} J^m(\boldsymbol{\theta}') - \nabla_{\theta_i} J^m(\boldsymbol{\theta})\|_1$$

$$= \frac{1}{1-\gamma^m} \sum_{s_i, a_i} \Big| d_{\boldsymbol{\rho}, i}^{\boldsymbol{\theta}', m}(s_i) \pi_{\theta_i'}(a_i|s_i) \overline{A_i^m}(s_i, a_i; \boldsymbol{\theta}')$$

$$- d_{\boldsymbol{\rho}, i}^{\boldsymbol{\theta}, m}(s_i) \pi_{\theta_i}(a_i|s_i) \overline{A_i^m}(s_i, a_i; \boldsymbol{\theta}) \Big|$$

$$\leq \frac{1}{1-\gamma^m} \sum_{s, a} |d_{\boldsymbol{\rho}}^{\boldsymbol{\theta}', m}(s) \boldsymbol{\pi}_{\boldsymbol{\theta}'}(a|s) A^m(s, a; \boldsymbol{\theta}')$$

$$- d_{\boldsymbol{\rho}}^{\boldsymbol{\theta}, m}(s) \boldsymbol{\pi}_{\boldsymbol{\theta}}(a|s) A^m(s, a; \boldsymbol{\theta})| \tag{45}$$

$$\leq \frac{1}{1-\gamma^m} \sum_{s, a} \Big( |d_{\boldsymbol{\rho}}^{\boldsymbol{\theta}', m}(s) \boldsymbol{\pi}_{\boldsymbol{\theta}'}(a|s)$$

$$- d_{\boldsymbol{\rho}}^{\boldsymbol{\theta}, m}(s) \boldsymbol{\pi}_{\boldsymbol{\theta}}(a|s)| \Big) A^m(s, a; \boldsymbol{\theta}')$$

$$+ d_{\boldsymbol{\rho}}^{\boldsymbol{\theta}, m}(s) \boldsymbol{\pi}_{\boldsymbol{\theta}}(a|s)|A^m(s, a; \boldsymbol{\theta}') - A^m(s, a; \boldsymbol{\theta})|$$

$$\leq \frac{1}{1-\gamma^m} \sum_{s, a} \frac{1}{1-\gamma^m} \big( |d_{\boldsymbol{\rho}}^{\boldsymbol{\theta}', m}(s) \boldsymbol{\pi}_{\boldsymbol{\theta}'}(a|s)$$

$$- d_{\boldsymbol{\rho}}^{\boldsymbol{\theta}, m}(s) \boldsymbol{\pi}_{\boldsymbol{\theta}}(a|s)| \big) + \max_{s, a} |A^m(s, a; \boldsymbol{\theta}') - A^m(s, a, \boldsymbol{\theta})|, \tag{46}$$

where the first inequality (45) can be obtained from the definition of $\overline{A_i^m}(s_i, a_i; \boldsymbol{\theta})$ in (39) and the fact that $|\sum_{i=1}^{N} x_i - \sum_{i=1}^{N} y_i| \leq \sum_{i=1}^{N} |x_i - y_i|, \forall x_i, y_i \in \mathbb{R}$, and the last inequality (46) comes from the fact that $A^m(s, a; \boldsymbol{\theta}) \leq 1/(1-\gamma^m)$.
For the right side of (46), we can use Corollary 35 and Lemma 32 in Zhang et al. (2022) to further obtain

$$\sum_{s} |d_{\boldsymbol{\rho}}^{\boldsymbol{\theta}', m}(s) \boldsymbol{\pi}_{\boldsymbol{\theta}'}(a|s) - d_{\boldsymbol{\rho}}^{\boldsymbol{\theta}, m}(s) \boldsymbol{\pi}_{\boldsymbol{\theta}}(a|s)|$$

$$\leq \frac{1}{1-\gamma^m} \max_{s} \|\boldsymbol{\pi}_{\boldsymbol{\theta}'}(\cdot|s) - \boldsymbol{\pi}_{\boldsymbol{\theta}}(\cdot|s)\|_1 \tag{47}$$

and

$$|A^m(s, a; \boldsymbol{\theta}') - A^m(s, a; \boldsymbol{\theta})|$$

$$\leq \frac{2}{(1-\gamma^m)^2} \max_{s} \|\boldsymbol{\pi}_{\boldsymbol{\theta}'}(\cdot|s) - \boldsymbol{\pi}_{\boldsymbol{\theta}}(\cdot|s)\|_1. \tag{48}$$

Substituting (47) and (48) into (46), we have

$$\|\nabla_{\theta_i} J^m(\boldsymbol{\theta}') - \nabla_{\theta_i} J^m(\boldsymbol{\theta})\|_1$$

$$\leq \frac{3}{(1-\gamma^m)^3} \max_{s} \|\boldsymbol{\pi}_{\boldsymbol{\theta}'}(\cdot|s) - \boldsymbol{\pi}_{\boldsymbol{\theta}}(\cdot|s)\|_1$$

$$= \frac{3}{(1-\gamma^m)^3} \sum_{i=1}^{N} \max_{s_i} \|\pi_{\theta_i'}(\cdot|s_i) - \pi_{\theta_i}(\cdot|s_i)\|_1 \tag{49}$$

$$\leq \frac{6}{(1-\gamma^m)^3} \sum_{i=1}^{N} \|\theta_i' - \theta_i\|_2, \tag{50}$$

where the last inequality (50) is obtained from Corollary 37 in Zhang et al. (2022) that for any two difference softmax policies $\pi_{\theta_i}$ and $\pi_{\theta_i'}$, and $s_i \in \mathcal{S}_i$, $\|\pi_{\theta_i}(\cdot|s_i) - \pi_{\theta_i'}(\cdot|s_i)\|_1 \leq 2\|\theta_i - \theta_i'\|_2$.

Combining (44) and (50), we further have

$$\|\nabla_{\boldsymbol{\theta}} J^m(\boldsymbol{\theta}) - \nabla_{\boldsymbol{\theta}} J^m(\boldsymbol{\theta}')\|_2^2$$

$$\leq \sum_{i=1}^N \Big( \frac{6}{(1-\gamma^m)^3} \sum_{i=1}^N \|\theta_i' - \theta_i\|_2 \Big)^2$$

$$= \frac{36N}{(1-\gamma^m)^6} \Big( \sum_{i=1}^N \|\theta_i' - \theta_i\|_2 \Big)^2$$

$$\leq \frac{36N^2}{(1-\gamma^m)^6} \sum_{i=1}^N \|\theta_i' - \theta_i\|_2^2 \tag{51}$$

$$\leq \frac{36N^2}{(1-\gamma^m)^6} \|\boldsymbol{\theta}' - \boldsymbol{\theta}\|_2^2, \tag{52}$$

where the second inequality (51) is obtained from that $\big( \sum_{i=1}^N x_i \big)^2 \leq N(\sum_{i=1}^N x_i^2), \forall x_i \in \mathbb{R}$.

According to $L_J = \max_{m \in \mathcal{M}} \frac{6N}{(1-\gamma^m)^3}$ and (52), we further have that $J^m(\boldsymbol{\theta})$ is $L_J$-smooth for all $m \in \mathcal{M}$.

## A.4 PROOF OF LEMMA 3

The detailed proof of Lemma 3 is provided in the following.

**Proof.** By the update of $g_{i,t}^m(b+1)$ in (13) and $g_{i,t}^m = g_{i,t}^m(B)$, we have

$$\nabla_{\theta_i} J^m(\boldsymbol{\theta}_t) - g_{i,t}^m$$

$$= \nabla_{\theta_i} J^m(\boldsymbol{\theta}_t) - \nabla_{\theta_i} J_{app,i}^m(\boldsymbol{\theta}_t) + \nabla_{\theta_i} J_{app,i}^m(\boldsymbol{\theta}_t) - g_{i,t}^m$$

$$= \nabla_{\theta_i} J^m(\boldsymbol{\theta}_t) - \nabla_{\theta_i} J_{app,i}^m(\boldsymbol{\theta}_t)$$

$$\quad + \sum_{h=0}^{\infty} (\gamma^m)^h \mathbb{E}\Big[ \nabla_{\theta_i} \log \pi_{\theta_{i,t}}(a_{i,h}|s_{i,h}) \widehat{Q_i^m}(\boldsymbol{s}_h, a_{i,h}; \boldsymbol{\theta}_t) \Big]$$

$$\quad - \frac{1}{B} \sum_{b=0}^{B-1} \sum_{h=0}^{H-1} (\gamma^m)^h \phi_i(s_{\mathcal{N}_i^\kappa, h}^b, a_{i,h}^b)^\top w_{i,t}$$

$$\quad \nabla_{\theta_i} \log \pi_{\theta_{i,t}}(a_{i,h}^b|s_{i,h}^b) \tag{53}$$

$$= \underbrace{\nabla_{\theta_i} J^m(\boldsymbol{\theta}_t) - \nabla_{\theta_i} J_{app,i}^m(\boldsymbol{\theta}_t)}_{\mathcal{T}_1}$$

$$\quad + \underbrace{\sum_{h=0}^{H-1} (\gamma^m)^h \mathbb{E}\Big[ \nabla_{\theta_i} \log \pi_{\theta_{i,t}}(a_{i,h}|s_{i,h}) \widehat{Q_i^m}(\boldsymbol{s}_h, a_{i,h}; \boldsymbol{\theta}_t) \Big]}_{\mathcal{T}_2}$$

$$\quad + \underbrace{\sum_{h=H}^{\infty} (\gamma^m)^h \mathbb{E}\Big[ \nabla_{\theta_i} \log \pi_{\theta_{i,t}}(a_{i,h}|s_{i,h}) \widehat{Q_i^m}(\boldsymbol{s}_h, a_{i,h}; \boldsymbol{\theta}_t) \Big]}_{\mathcal{T}_3}$$

$$\quad - \underbrace{\frac{1}{B} \sum_{b=0}^{B-1} \sum_{h=0}^{H-1} (\gamma^m)^h \nabla_{\theta_i} \log \pi_{\theta_{i,t}}(a_{i,h}^b|s_{i,h}^b) \widehat{Q_i^m}(s_h^b, a_{i,h}^b; \boldsymbol{\theta}_t)}_{\mathcal{T}_4}$$

$$\quad + \underbrace{\frac{1}{B} \sum_{b=0}^{B-1} \sum_{h=0}^{H-1} (\gamma^m)^h \nabla_{\theta_i} \log \pi_{\theta_{i,t}}(a_{i,h}^b|s_{i,h}^b)}_{\mathcal{T}_5}$$

$$\underbrace{\left(\widehat{Q_i^m}(s_h^b, a_{i,h}^b; \boldsymbol{\theta}_t) - \phi_i(s_{\mathcal{N}_i^\kappa, h}^b, a_{i,h}^b)^\top w_{i,t}\right)}_{\mathcal{T}_5}, \tag{54}$$

where the equality (53) can be obtained by the policy gradient theorem variant (i.e., Lemma F.1 in Zhou et al. (2023)). Based on (54), we have

$$\mathbb{E}[\|\nabla_{\theta_i} J^m(\boldsymbol{\theta}_t) - g_{i,t}^m\|_2^2]$$
$$= \mathbb{E}[\|\mathcal{T}_1 + \mathcal{T}_2 + \mathcal{T}_3 - \mathcal{T}_4 + \mathcal{T}_5\|_2^2]$$
$$\leq 4\mathbb{E}[\|\mathcal{T}_1\|_2^2 + \|\mathcal{T}_2 - \mathcal{T}_4\|_2^2 + \|\mathcal{T}_3\|_2^2 + \|\mathcal{T}_5\|_2^2]. \tag{55}$$
$$\leq \frac{8R^2}{(1-\gamma^m)^4}(\gamma^m)^{2\kappa+2} + \frac{32}{(1-\gamma^m)^2 B} + \frac{8(\gamma^m)^{2H}}{(1-\gamma^m)^4}$$
$$+ \frac{8\varepsilon_{critic}^{\boldsymbol{\theta}_t}}{(1-\gamma^m)^2}. \tag{56}$$

where (56) can be obtained by (35) and the definition of $\varepsilon_{critic}^{\boldsymbol{\theta}_t}$ in (20). $\qquad\square$

## A.5 PROOF OF 1

**Proof.** Based on (Olfati-Saber & Murray, 2004), the results of (i) and (ii) are straightforward; therefore, we will focus on proving (iii). Since $J_t^g$ is $L_t^g$-Lipschitz continuous and $\boldsymbol{\lambda}_i(k) = \boldsymbol{\lambda}_j(k)$ for all $i, j \in \mathcal{N}$, we define $\boldsymbol{\lambda}(k) = \boldsymbol{\lambda}_i(k)$ and have

$$J_t^g\big(\boldsymbol{\lambda}(k+1)\big) \leq J_t^g\big(\boldsymbol{\lambda}(k)\big) + \alpha_k\big\langle\nabla J_t^g\big(\boldsymbol{\lambda}(k)\big), \boldsymbol{e}_{u_{i,t}^*(k)} - \boldsymbol{\lambda}(k)\big\rangle$$
$$+ \frac{L_t^g}{2}\alpha_k^2\|\boldsymbol{e}_{u_{i,t}^*(k)} - \boldsymbol{\lambda}(k)\|^2$$
$$\leq J_t^g\big(\boldsymbol{\lambda}(k)\big) + \alpha_k\big\langle\nabla J_t^g\big(\boldsymbol{\lambda}(k)\big), \boldsymbol{e}_{u_{i,t}^*(k)} - \boldsymbol{\lambda}(k)\big\rangle$$
$$+ \alpha_k^2 L_t^g, \tag{57}$$

where the last inequality can be obtained by the fact that $\|\boldsymbol{e}_{u_{i,t}^*(k)} - \boldsymbol{\lambda}(k)\| \leq \sqrt{2}$. Since $u_{i,t}^*(k) = \arg\min_m y_{i,t}^m(k)$, we can obtain that

$$\big\langle\nabla J_t^g\big(\boldsymbol{\lambda}(k)\big), \boldsymbol{e}_{u_{i,t}^*(k)} - \boldsymbol{\lambda}(k)\big\rangle \leq \big\langle\nabla J_t^g\big(\boldsymbol{\lambda}(k)\big), \boldsymbol{\lambda}_t^* - \boldsymbol{\lambda}(k)\big\rangle$$
$$\leq J_t^g(\boldsymbol{\lambda}_t^*) - f\big(\boldsymbol{\lambda}(k)\big). \tag{58}$$

Combining (58) with (57), we have

$$J_t^g\big(\boldsymbol{\lambda}(k+1)\big) - J_t^g(\boldsymbol{\lambda}_t^*) \leq (1 - \alpha_k)\Big(J_t^g\big(\boldsymbol{\lambda}(k)\big) - J_t^g(\boldsymbol{\lambda}_t^*)\Big)$$
$$+ \alpha_k^2 L_t^g. \tag{59}$$

By induction, (59) implies that $J_t^g\big(\boldsymbol{\lambda}(k+1)\big) - J_t^g(\boldsymbol{\lambda}_t^*) \leq \frac{4L_t^g}{k+1}$. By the fact that $\widehat{\boldsymbol{\lambda}}_t = \boldsymbol{\lambda}(K_\lambda)$, we can obtain $J_t^g(\widehat{\boldsymbol{\lambda}}_t) - J_t^g(\boldsymbol{\lambda}_t^*) \leq \frac{4L_t^g}{K_\lambda+1}$. $\qquad\square$

## A.6 PROOF OF THEOREM 2

**Proof.** According to the smoothness of $J^m(\boldsymbol{\theta})$ in Lemma 2 and $L_J = \max_{m\in\mathcal{M}} \frac{6N}{(1-\gamma^m)^3}$, we can have

$$J^m(\boldsymbol{\theta}_{t+1}) \geq J^m(\boldsymbol{\theta}_t) + \langle\nabla_{\boldsymbol{\theta}} J^m(\boldsymbol{\theta}_t), \boldsymbol{\theta}_{t+1} - \boldsymbol{\theta}_t\rangle$$
$$- \frac{L_J}{2}\|\boldsymbol{\theta}_{t+1} - \boldsymbol{\theta}_t\|_2^2, \forall m \in \mathcal{M}. \tag{60}$$

Taking $\boldsymbol{\lambda}_t$ weighted summation over (60), we have

$$\boldsymbol{\lambda}_t^\top \boldsymbol{J}(\boldsymbol{\theta}_{t+1})$$
$$\geq \boldsymbol{\lambda}_t^\top \boldsymbol{J}(\boldsymbol{\theta}_t) + \langle\nabla_{\boldsymbol{\theta}} \boldsymbol{J}(\boldsymbol{\theta}_t)^\top \boldsymbol{\lambda}_t, \boldsymbol{\theta}_{t+1} - \boldsymbol{\theta}_t\rangle$$
$$- \frac{L_J}{2}\|\boldsymbol{\theta}_{t+1} - \boldsymbol{\theta}_t\|_2^2$$
$$= \boldsymbol{\lambda}_t^\top \boldsymbol{J}(\boldsymbol{\theta}_t) + \eta_{\boldsymbol{\theta},t}\Big\langle\nabla_{\boldsymbol{\theta}} \boldsymbol{J}(\boldsymbol{\theta}_t)^\top \boldsymbol{\lambda}_t, \sum_{m=1}^M \lambda_t^m \boldsymbol{g}_t^m\Big\rangle$$

$$- \frac{L_J \eta_{\boldsymbol{\theta},t}^2}{2} \Big\| \sum_{m=1}^{M} \lambda_t^m \boldsymbol{g}_t^m \Big\|_2^2 \tag{61}$$

$$= \boldsymbol{\lambda}_t^\top \boldsymbol{J}(\boldsymbol{\theta}_t) + \eta_{\boldsymbol{\theta},t} \Big\langle \nabla_{\boldsymbol{\theta}} \boldsymbol{J}(\boldsymbol{\theta}_t)^\top \boldsymbol{\lambda}_t, $$

$$\sum_{m=1}^{M} \lambda_t^m \big( \boldsymbol{g}_t^m - \nabla_{\boldsymbol{\theta}} J^m(\boldsymbol{\theta}_t) + \nabla_{\boldsymbol{\theta}} J^m(\boldsymbol{\theta}_t) \big) \Big\rangle$$

$$- \frac{L_J \eta_{\boldsymbol{\theta},t}^2}{2} \Big\| \sum_{m=1}^{M} \lambda_t^m \boldsymbol{g}_t^m \Big\|_2^2$$

$$= \boldsymbol{\lambda}_t^\top \boldsymbol{J}(\boldsymbol{\theta}_t) + \eta_{\boldsymbol{\theta},t} \Big\langle \nabla_{\boldsymbol{\theta}} \boldsymbol{J}(\boldsymbol{\theta}_t)^\top \boldsymbol{\lambda}_t, \sum_{m=1}^{M} \lambda_t^m \nabla_{\boldsymbol{\theta}} J^m(\boldsymbol{\theta}_t) \Big\rangle$$

$$+ \eta_{\boldsymbol{\theta},t} \Big\langle \nabla_{\boldsymbol{\theta}} \boldsymbol{J}(\boldsymbol{\theta}_t)^\top \boldsymbol{\lambda}_t, \sum_{m=1}^{M} \lambda_t^m \big( \boldsymbol{g}_t^m$$

$$- \nabla_{\boldsymbol{\theta}} J^m(\boldsymbol{\theta}_t) \big) \Big\rangle - \frac{L_J \eta_{\boldsymbol{\theta},t}^2}{2} \Big\| \sum_{m=1}^{M} \lambda_t^m \boldsymbol{g}_t^m \Big\|_2^2$$

$$\geq \boldsymbol{\lambda}_t^\top \boldsymbol{J}(\boldsymbol{\theta}_t) + \frac{\eta_{\boldsymbol{\theta},t}}{2} \| \nabla_{\boldsymbol{\theta}} \boldsymbol{J}(\boldsymbol{\theta}_t)^\top \boldsymbol{\lambda}_t \|_2^2$$

$$- \frac{\eta_{\boldsymbol{\theta},t}}{2} \Big\| \sum_{m=1}^{M} \lambda_t^m \big( \boldsymbol{g}_t^m - \nabla_{\boldsymbol{\theta}} J^m(\boldsymbol{\theta}_t) \big) \Big\|_2^2$$

$$- \frac{L_J \eta_{\boldsymbol{\theta},t}^2}{2} \Big\| \sum_{m=1}^{M} \lambda_t^m \big( \boldsymbol{g}_t^m - \nabla_{\boldsymbol{\theta}} J^m(\boldsymbol{\theta}_t) + \nabla_{\boldsymbol{\theta}} J^m(\boldsymbol{\theta}_t) \big) \Big\|_2^2 \tag{62}$$

$$\geq \boldsymbol{\lambda}_t^\top \boldsymbol{J}(\boldsymbol{\theta}_t) + \Big( \frac{\eta_{\boldsymbol{\theta},t}}{2} - L_J \eta_{\boldsymbol{\theta},t}^2 \Big) \| \nabla_{\boldsymbol{\theta}} \boldsymbol{J}(\boldsymbol{\theta}_t)^\top \boldsymbol{\lambda}_t \|_2^2$$

$$- \Big( \frac{\eta_{\boldsymbol{\theta},t}}{2} + L_J \eta_{\boldsymbol{\theta},t}^2 \Big) \Big\| \sum_{m=1}^{M} \lambda_t^m \big( \boldsymbol{g}_t^m - \nabla_{\boldsymbol{\theta}} J^m(\boldsymbol{\theta}_t) \big) \Big\|_2^2, \tag{63}$$

where the equality (61) comes from (18), the inequality (62) can be obtained by the fact that $\langle x, y \rangle \geq -\frac{1}{2}(\|x\|^2 + \|y\|^2), \forall x, y \in \mathbb{R}^{\sum_{i=1}^{N} |\mathcal{S}_i||\mathcal{A}_i|}$, and the inequality (63) can be get by the fact that $\|x + y\|_2^2 \leq 2(\|x\|_2^2 + \|y\|_2^2), \forall x, y \in \mathbb{R}^{\sum_{i=1}^{N} |\mathcal{S}_i||\mathcal{A}_i|}$. By (63), we have

$$\| \nabla_{\boldsymbol{\theta}} \boldsymbol{J}(\boldsymbol{\theta}_t)^\top \boldsymbol{\lambda}_t \|_2^2$$

$$\leq \frac{2 \big( \boldsymbol{\lambda}_t^\top \boldsymbol{J}(\boldsymbol{\theta}_{t+1}) - \boldsymbol{\lambda}_t^\top \boldsymbol{J}(\boldsymbol{\theta}_t) \big)}{\eta_{\boldsymbol{\theta},t} - 2 \eta_{\boldsymbol{\theta},t}^2 L_J}$$

$$+ \frac{\eta_{\boldsymbol{\theta},t} + 2 \eta_{\boldsymbol{\theta},t}^2 L_J}{\eta_{\boldsymbol{\theta},t} - 2 \eta_{\boldsymbol{\theta},t}^2 L_J} \Big\| \sum_{m=1}^{M} \lambda_t^m \big( \nabla_{\boldsymbol{\theta}} J^m(\boldsymbol{\theta}_t) - \boldsymbol{g}_t^m \big) \Big\|_2^2. \tag{64}$$

Consider that $\widehat{\boldsymbol{\lambda}}_t$ is the optimal of problem (15), we have

$$\| \nabla_{\boldsymbol{\theta}} \boldsymbol{J}(\boldsymbol{\theta}_t)^\top \widehat{\boldsymbol{\lambda}}_t \|_2^2 \leq \| \nabla_{\boldsymbol{\theta}} \boldsymbol{J}(\boldsymbol{\theta}_t)^\top \boldsymbol{\lambda}_t \|_2^2 + \frac{4 L_t^g}{K_\lambda + 1}. \tag{65}$$

Using the setting of the learning rate as $\eta_{\boldsymbol{\theta},t} = \frac{1}{3 L_J}$ and taking expectation on both side of (64), we further have

$$\mathbb{E}[\| \nabla_{\boldsymbol{\theta}} \boldsymbol{J}(\boldsymbol{\theta}_t)^\top \widehat{\boldsymbol{\lambda}}_t \|_2^2]$$

$$\leq 18 L_J \mathbb{E}[\boldsymbol{\lambda}_t^\top \boldsymbol{J}(\boldsymbol{\theta}_{t+1}) - \boldsymbol{\lambda}_t^\top \boldsymbol{J}(\boldsymbol{\theta}_t)] + \mathbb{E}\Big[ \frac{4 L_t^g}{K_\lambda + 1} \Big]$$

$$+ 5 \mathbb{E}\Big[ \sum_{m=1}^{M} \lambda_t^m \| \nabla_{\boldsymbol{\theta}} J^m(\boldsymbol{\theta}_t) - \boldsymbol{g}_t^m \|_2^2 \Big]$$

$$\leq 18 L_J \mathbb{E}[\boldsymbol{\lambda}_t^\top \boldsymbol{J}(\boldsymbol{\theta}_{t+1}) - \boldsymbol{\lambda}_t^\top \boldsymbol{J}(\boldsymbol{\theta}_t)]$$

$$+ 5 \max_{m \in \mathcal{M}} \varepsilon_{actor}^m + \frac{8}{K_\lambda + 1} \Big( \max_{m \in \mathcal{M}} (\varepsilon_{actor}^m)^2 + \max_{m \in \mathcal{M}} \frac{2R^2}{(1 - \gamma^m)^4} \Big), \tag{66}$$

where the last inequality comes from Lemma 3. Taking average of (66) over $T$, we have

$$\frac{1}{T} \sum_{t=1}^{T} \mathbb{E}[\|\nabla_{\boldsymbol{\theta}} \boldsymbol{J}(\boldsymbol{\theta}_t)^\top \widehat{\boldsymbol{\lambda}}_t\|_2^2]$$

$$\leq \frac{1}{T} \sum_{t=1}^{T} 18 L_J \mathbb{E}[\boldsymbol{\lambda}_t^\top \boldsymbol{J}(\boldsymbol{\theta}_{t+1}) - \boldsymbol{\lambda}_t^\top \boldsymbol{J}(\boldsymbol{\theta}_t)]$$

$$+ 5 \max_{m \in \mathcal{M}} \varepsilon_{actor}^m + \frac{8}{K_\lambda + 1} \Big( \max_{m \in \mathcal{M}} (\varepsilon_{actor}^m)^2 + \max_{m \in \mathcal{M}} \frac{2R^2}{(1 - \gamma^m)^4} \Big). \tag{67}$$

Considering that

$$\sum_{t=1}^{T} \mathbb{E}[\boldsymbol{\lambda}_t^\top \boldsymbol{J}(\boldsymbol{\theta}_{t+1}) - \boldsymbol{\lambda}_t^\top \boldsymbol{J}(\boldsymbol{\theta}_t)]$$

$$= \mathbb{E}\Big[ \sum_{t=1}^{T-1} (-\boldsymbol{\lambda}_{t+1} + \boldsymbol{\lambda}_t)^\top \boldsymbol{J}(\boldsymbol{\theta}_{t+1}) - \boldsymbol{\lambda}_1^\top \boldsymbol{J}(\boldsymbol{\theta}_1)$$

$$+ \boldsymbol{\lambda}_T^\top \boldsymbol{J}(\boldsymbol{\theta}_{T+1}) \Big]$$

$$\leq \mathbb{E}\Big[ \sum_{t=1}^{T-1} \| - \boldsymbol{\lambda}_{t+1} + \boldsymbol{\lambda}_t \|_1 \| \boldsymbol{J}(\boldsymbol{\theta}_{t+1}) \|_\infty + |\boldsymbol{\lambda}_1|_1 \| \boldsymbol{J}(\boldsymbol{\theta}_1) \|_\infty$$

$$+ |\boldsymbol{\lambda}_T|_\infty \| \boldsymbol{J}(\boldsymbol{\theta}_{T+1}) \|_\infty \Big] \tag{68}$$

$$\leq \sum_{t=1}^{T-1} \Big[ \eta_{\boldsymbol{\lambda},t} \mathbb{E}[\| \boldsymbol{\lambda}_t - \widehat{\boldsymbol{\lambda}}_t \|_1] \frac{1}{1 - \|\boldsymbol{\gamma}\|_\infty} \Big] + \frac{2}{1 - \|\boldsymbol{\gamma}\|_\infty} \tag{69}$$

$$\leq \frac{2}{1 - \|\boldsymbol{\gamma}\|_\infty} \Big( 1 + \sum_{t=1}^{T} \eta_{\boldsymbol{\lambda},t} \Big), \tag{70}$$

where the inequality (68) comes from the fact that $x^\top y \leq \|x\|_1 \|y\|_\infty, \forall x, y \in \mathbb{R}^M$ and the inequality (69) is obtained from the update of $\boldsymbol{\lambda}_t$ in (17). Taking (70) into (67), we can prove this theorem. $\square$