# OpenReview forum: "Distributed Algorithm for Multi-objective Multi-agent Reinforcement Learning"
_ICLR.cc/2026/Conference — Submitted to ICLR 2026_

### Official Review · Reviewer_LYAQ · 2025-10-31

**Soundness:** 3
**Presentation:** 2
**Contribution:** 1
**Rating:** 2
**Confidence:** 4

**Summary:**

This paper tackles the challenges of high computational complexity and limited communication in large-scale distributed multi-objective multi-agent reinforcement learning (MOMARL) by proposing a distributed algorithm. The approach introduces an approximate policy gradient and linear function approximation based on local neighborhood information to effectively reduce the state–action dimensionality, and employs a consensus-based protocol to adaptively adjust multi-objective weights, enabling convergence to an approximate Pareto-equilibrium solution using only local communication. Theoretical analysis proves that the algorithm achieves an O(1/T) convergence rate, and experiments on a multi-robot path planning task validate its effectiveness.

**Strengths:**

1. The problem addressed in this paper is clearly defined, and the motivation is well articulated.
2. A distributed and scalable algorithm is proposed under the MOMARL framework, demonstrating originality.
3. The algorithm design is reasonable, and the overall description is clear.

**Weaknesses:**

1. The literature review is insufficient: the introduction provides an inadequate discussion of existing MOMARL and distributed RL algorithms, lacking direct comparison with representative works; moreover, the references are relatively outdated and the related work section is missing.

2. The experimental setup is not clearly described, with insufficient details regarding the design of the neighborhood network and other implementation aspects; it is recommended to include comparisons with baseline methods from recent MOMARL studies.

**Questions:**

1. The paper does not review relevant studies on fully decentralized multi-agent reinforcement learning (MARL). Excluding the multi-objective component, it remains unclear how the proposed method fundamentally differs from existing fully decentralized MARL approaches. The paper lacks a Related Work section to situate its contribution in the broader literature.

2. From the current description, it appears that each agent’s local state space includes information from all nodes, which contradicts the notion of locality. The experimental section should provide more detailed explanations of the experimental design, particularly clarifying how the k-hop neighborhood is defined and implemented.

3. Although the paper claims to develop a fully distributed algorithm, the definition of the Q-function indicates dependence on neighboring agents’ rewards. The authors should clarify what “fully distributed” means in the framework—whether it allows partial information exchange among neighbors or strictly prohibits it.

4. The description of the feature vector mapping is insufficient. It is unclear whether this mapping is a user-defined linear function and whether it serves to restrict which neighbors are included for agent $i$. A more explicit formulation or example would improve clarity.

5. In the second term of Equation (12), the normalization factor should likely be the size of the neighborhood $|N_i|$ rather than the total number of agents $N$. This correction is important for maintaining consistency and correct scaling in the local TD update.

---

> ### Author Response · Authors · 2025-12-02
>
> W1. Most recently, extensive research has been conducted in the field of single-agent multi-objective reinforcement learning [R1]-[R8], but a few approaches can solve the multi-agent multi-objective reinforcement learning [R9]-[R11]. However, these work require  the global stat-action information. In our work, each agent only needs its $\kappa$-hop neighbors' state information.
>
> [R1] Multi-objective reinforcement learning using sets of Pareto dominating policies.
> [R2] Pareto-DQN: Approximating the Pareto front in complex multi-objective decision problems.
> [R3] Intrinsically motivated hierarchical policy learning in multi objective Markov decision processes.
> [R4] Multi objective deep reinforcement learning.
> [R5] Dynamic weights in multi-objective deep reinforcement learning.
> [R6] Meta-learning for multi-objective reinforcement learning.
> [R7] Prediction-guided multi-objective reinforcement learning for continuous robot control.
> [R8] A generalized algorithm for multi-objective reinforcement learning and policy adaptation.
> [R9] MO-MIX: Multi-objective multi-agent cooperative decision-making with deep reinforcement learning.
> [R10] Momaland: A set of benchmarks for multi-objective multi-agent reinforcement learning.
> [R11] MOMA-AC: A preference-driven actor-critic framework for continuous multi-objective multi-agent reinforcement learning.
>
> W2. We have added a comprehensive ablation studies to isolate the effects of key components of our method, including the influence of $kappa$-hop locality.
>
> Q1.  Existing fully decentralized MARL approaches typically decentralize the action-selection process but still require each agent to access global state information [R12]-[R15].
> In contrast, our framework is fundamentally different: each agent relies solely on the state information of its $\kappa$-hop neighbors and its own local action information, without requiring any global state.
>
> [R12] MA2QL: A minimalist approach to fully decentralized multi-agent reinforcement learning.
> [R13] Sample and Communication Efficient Fully Decentralized MARL Policy Evaluation via a New Approach: Local TD Update.
> [R14] Fully decentralized multi-agent reinforcement learning with networked agents.
> [R15] Finite-sample analysis for decentralized batch multiagent reinforcement learning with networked agents.
>
> Q2. The $\kappa$-hop neighbor of agent $i$ is defined as the set of the agents whose graph distance to $i$ is less than or equal to $\kappa$, including i itself.
>
> Q3. In our distributed setting, each agent is allowed to access information from its $\kappa$-hop neighbors, including states and rewards.
>
> Q4. In our simulations, the feature vector for agent $i$ is defined as a binary vector of length $|\mathcal{S}\_{\mathcal{N}^{\kappa}\_{i}}||\mathcal{A}\_{i}|$. Specifically, the element corresponding to the joint state-action pair $(s\_{\mathcal{N}^{\kappa}\_{i}}, a\_i)$ is set to $1$, while all other elements are set to $0$.
>
> Q5. Eq. (12) is closely related to Eq. (7), which forms the core of our analysis. As such, the normalization by the global factor $N$ is intentional and necessary for theoretical consistency. Using the local neighborhood size $|\mathcal{N}^{\kappa}_{i}|$ instead would introduce an additional bias/error term $\frac{\sqrt{2}(N-|\mathcal{N}^{\kappa}\_{i}|)R}{(1-\gamma^{m})^{2}|\mathcal{N}^{\kappa}\_{i}|}$ in Theorem 1 that is not captured by our current theoretical framework.

---

### Official Review · Reviewer_e4Vc · 2025-11-01

**Soundness:** 2
**Presentation:** 2
**Contribution:** 2
**Rating:** 4
**Confidence:** 2

**Summary:**

The paper proposes a fully distributed algorithm for multi-objective multi-agent reinforcement learning, where each agent updates its policy using a localized, action-averaged critic over its κ-hop neighborhood and negotiates objective trade-offs via a consensus–Frank–Wolfe update of weights. The authors derive bounds showing that the policy-gradient approximation error decays with neighborhood radius and prove convergence to an $\varepsilon$-Pareto-stationary point under standard assumptions. Experiments on multi-robot path planning report higher per-objective returns and faster optimization (smaller $\|g_t\|_2$) than a centralized MORL baseline.

**Strengths:**

The paper offers a principled localization of policy gradients to κ-hop neighborhoods with a clear approximation bound that decays as $(\gamma)^{\kappa+1}$ and a proof of convergence to an $\varepsilon$-Pareto-stationary point. Its communication-efficient consensus–Frank–Wolfe weighting—using only neighbor information—translates to practical multi-robot gains, showing higher per-objective returns and faster reduction of $\|g_t\|_2$ than a centralized baseline.

**Weaknesses:**

- Consensus implementation vs. theory mismatch: Algorithm 2 uses “while-until-exact-consensus,” implying unbounded communication, whereas the analysis assumes a fixed $K_\lambda$; the bounds omit an explicit consensus residual, so finite-round practice either violates assumptions or undermines scalability.
- Core approximation issues: the gradient-error bound lacks explicit dependence on graph topology (degree, spectral gap, neighborhood size), and the critic/TD scaling sums local rewards but divides by global $N$; when $|\mathcal N_i^\kappa|\!\ll\! N$, this compresses magnitudes, distorts variance, and complicates step-size selection (a normalization by $|\mathcal N_i^\kappa|$ would be more coherent).
- Weak empirical support for key claims: only a single centralized baseline is considered, with no Pareto frontier, significance tests, ablations, or communication/runtime reporting; claims of being “near central optimum” and “faster” lack a verifiable upper bound or comparisons to strong MARL baselines.

**Questions:**

- Could you broaden the empirical study to plot full Pareto frontiers (by sweeping initial weights), compare against strong MARL baselines (e.g., VDN, QMIX, MAPPO, MADDPG), report mean±std over multiple seeds with significance tests, and quantify communication per update and wall-clock time relative to a centralized approach?

---

> ### Author Response · Authors · 2025-11-19
>
> W1. Most recently, extensive research has been conducted in the field of single-agent multi-objective reinforcement learning [R1]-[R8], but a few approaches can solve the multi-agent multi-objective reinforcement learning [R9]-[R11]. However, these work require  the global stat-action information. In our work, each agent only needs its $\kappa$-hop neighbors' state information.
>
> [R1] Multi-objective reinforcement learning using sets of Pareto dominating policies.
> [R2] Pareto-DQN: Approximating the Pareto front in complex multi-objective decision problems.
> [R3] Intrinsically motivated hierarchical policy learning in multi objective Markov decision processes.
> [R4] Multi objective deep reinforcement learning.
> [R5] Dynamic weights in multi-objective deep reinforcement learning.
> [R6] Meta-learning for multi-objective reinforcement learning.
> [R7] Prediction-guided multi-objective reinforcement learning for continuous robot control.
> [R8] A generalized algorithm for multi-objective reinforcement learning and policy adaptation.
> [R9] MO-MIX: Multi-objective multi-agent cooperative decision-making with deep reinforcement learning.
> [R10] Momaland: A set of benchmarks for multi-objective multi-agent reinforcement learning.
> [R11] MOMA-AC: A preference-driven actor-critic framework for continuous multi-objective multi-agent reinforcement learning.
>
> W2. We have added a comprehensive ablation studies to isolate the effects of key components of our method, including the influence of $kappa$-hop locality.
>
> Q1.  Existing fully decentralized MARL approaches typically decentralize the action-selection process but still require each agent to access global state information [R12]-[R15].
> In contrast, our framework is fundamentally different: each agent relies solely on the state information of its $\kappa$-hop neighbors and its own local action information, without requiring any global state.
>
> [R12] MA2QL: A minimalist approach to fully decentralized multi-agent reinforcement learning.
> [R13] Sample and Communication Efficient Fully Decentralized MARL Policy Evaluation via a New Approach: Local TD Update.
> [R14] Fully decentralized multi-agent reinforcement learning with networked agents.
> [R15] Finite-sample analysis for decentralized batch multiagent reinforcement learning with networked agents.
>
> Q2. The $\kappa$-hop neighbor of agent $i$ is defined as the set of the agents whose graph distance to $i$ is less than or equal to $\kappa$, including i itself.
>
> Q3. In our distributed setting, each agent is allowed to access information from its $\kappa$-hop neighbors, including states and rewards.
>
> Q4. In our simulations, the feature vector for agent $i$ is defined as a binary vector of length $|\mathcal{S}\_{\mathcal{N}^{\kappa}\_{i}}||\mathcal{A}\_{i}|$. Specifically, the element corresponding to the joint state-action pair $(s\_{\mathcal{N}^{\kappa}\_{i}}, a\_i)$ is set to $1$, while all other elements are set to $0$.
>
> Q5. Eq. (12) is closely related to Eq. (7), which forms the core of our analysis. As such, the normalization by the global factor $N$ is intentional and necessary for theoretical consistency. Using the local neighborhood size $|\mathcal{N}^{\kappa}_{i}|$ instead would introduce an additional bias/error term $\frac{\sqrt{2}(N-|\mathcal{N}^{\kappa}\_{i}|)R}{(1-\gamma^{m})^{2}|\mathcal{N}^{\kappa}\_{i}|}$ in Theorem 1 that is not captured by our current theoretical framework.

---

> > ### Comment · Reviewer_e4Vc · 2025-11-27
> >
> > The authors' rebuttal addresses some of my concerns, and I have decided to maintain the original score.

---

### Official Review · Reviewer_atGW · 2025-11-03

**Soundness:** 3
**Presentation:** 3
**Contribution:** 3
**Rating:** 6
**Confidence:** 4

**Summary:**

The paper presents a fully distributed actor-critic algorithm for multi-objective multi-agent reinforcement learning. Each agent uses only $\kappa$-hop neighborhood states and its local action, together with an actionaveraged $Q$-function and linear function approximation, to estimate policy gradients. A consensus-plus-Frank-Wolfe procedure adjusts objective weights. The authors prove $O(1 / T)$ convergence to an $\varepsilon$-Pareto-stationary solution under standard assumptions on rewards and network connectivity conditions.

**Strengths:**

Strengths:


1. The paper replaces the global policy gradient with a local, action-averaged gradient $\nabla_{\theta_i} J_{\text {app }, i}^m(\theta)$ that depends only on the agent's own action and $\kappa$-hop neighborhood. It then proves a geometrically decaying approximation error
$\left\|\nabla_{\theta_i} J_{\mathrm{app}, i}^m(\theta)-\nabla_{\theta_i} J^m(\theta)\right\|_2 \leq \frac{\sqrt{2 R}}{\left(1-\gamma^m\right)^2}\left(\gamma^m\right)^{\kappa+1}$
which makes the approximation transparent and parameterized by $\kappa$.


2. The critic uses a linear approximation of the form
$\hat{Q}^m_i\left(s_{N_i^\kappa}, a_i ; w_i^m\right)=\phi_i\left(s_{N_i^\kappa}, a_i\right)^{\top} w_i^m$
so that each agent only estimates values over its neighborhood state and its own action. This aligns the value approximation with the locality assumption used in the policy-gradient derivation.

3. Distributed weight selection that preserves the convergence rate.
The scalarization/weight-selection step is solved by a consensus plus Frank-Wolfe update over the network. The analysis shows that, despite decentralization, the overall algorithm still achieves an $O(1 / T)$ convergence rate to an $\varepsilon$-Pareto-stationary point, which is nontrivial for multi-objective, multi-agent settings.

**Weaknesses:**

Weaknesses:

1. The final actor update bundles truncation, sampling, and function-approximation errors into a single term; the paper shows it stays bounded, but does not give a tight characterization of how this bound scales with all inner-loop parameters, so the sharpness of the $O(1 / T)$ claim is partially opaque.

2. The key bound decays as $\left(\gamma^m\right)^{\kappa+1} /\left(1-\gamma^m\right)^2$. When $\gamma^m$ is close to 1 and $\kappa$ must stay small for communication reasons, this term can be large, so the locality that makes the method scalable can simultaneously weaken the approximation guarantee.

3. The distributed weight-update step requires (at least approximate) consensus at every outer iteration. The theory assumes this is done sufficiently well, but the per-iteration communication/synchronization burden is not incorporated into the main convergence complexity statement.

**Questions:**

1. Normalization in critic updates: The TD-style critic uses localized information. Is the global normalization factor (or averaging step) essential for the contraction argument, or could a purely local normalization reduce variance without breaking the proof?

2. The Frank-Wolfe-based weight update is analyzed as if the subproblem is solved well each round. How does the main convergence bound change if only a fixed, small number of FW steps (and consensus rounds) is used per iteration?

3. The policy-gradient derivation is presented for the softmax/discrete case. Can the same local-action, action-averaged construction be extended to deterministic or Gaussian policies in continuous action spaces while still retaining the $O(1 / T)$ rate?

---

> ### Author Response · Authors · 2025-11-19
>
> W1. The consensus requirement in the distributed weight-update step can indeed be achieved within a finite number of iterations. See the finite-time consensus results in ``Finite-time distributed consensus in graphs with time-invariant topologies.'' Algorithm 2 builds on this consensus mechanism to compute an approximate solution to problem (15).
>
> W2. Eqs (7)-(9) constitute the core technical construction of our method, and the normalization by the global factor $N$ is essential for our analysis. Using the local neighborhood size $|\mathcal{N}^{\kappa}_{i}|$ instead would introduce an additional bias/error term $\frac{\sqrt{2}(N-|\mathcal{N}^{\kappa}\_{i}|)R}{(1-\gamma^{m})^{2}|\mathcal{N}^{\kappa}\_{i}|}$ in Theorem 1 that is not captured by our current theoretical framework, because the local normalization would misalign the magnitude of aggregated TD/residual signals relative to the global objective.
>
> W3. We have added a comprehensive ablation studies to isolate the effects of key components of our method, including the influence of $\kappa$-hop locality.
>
> Q1. We note that these methods are specifically designed to address cooperative multi-agent optimization problems, rather than a multi-objective multi-agent optimization problem. Therefore, the problem settings are inherently different, and a direct comparison is not meaningful.
>
> We hope that the clarifications provided above adequately address the reviewer’s concerns, and we would be grateful if these revisions could be taken into account in a more favorable reassessment of the manuscript’s rating.

---

### Author Response · Authors · 2025-11-19

W1. In fact, Eq. (21) provides an explicit characterization of the policy-gradient approximation error, showing how it depends on the truncation distance, the sample length, and the linear function-approximation. This expression clarifies the source of the aggregated error term and its scaling behavior with respect to the inner-loop parameters.

W2. This term arises from the stage-wise decomposition technique that enables the scalable structure of our method, and similar decay factors appear in prior works such as~``Scalable reinforcement learning of localized policies for multi-agent networked systems".
Moreover, in our simulations we set $\kappa = 1$ and $\gamma = 0.9$. Although the resulting policy-gradient estimate exhibits a nonzero gap from zero, the empirical results consistently show that our algorithm significantly outperforms existing baselines under these practical parameter choices.

W3. The consensus requirement in the distributed weight-update step can indeed be achieved within a finite number of iterations. See the finite-time consensus results in ``Finite-time distributed consensus in graphs with time-invariant topologies.'' Algorithm 2 builds on this consensus mechanism to compute an approximate solution to problem (15). Therefore, the communication cost associated with achieving consensus does not need to be incorporated into the main convergence complexity, as it can be treated as a separate finite-time subroutine rather than a per-iteration burden of the optimization process.

Q1. Yes, the key fixed-point operator used in the critic analysis contracts only after aggregating all local TD residuals into a globally normalized quantity; without this step, the disagreement across agents introduces additional bias terms that prevent the operator from being contractive in the required norm.

Q2. Inspired by the finite-time consensus results in ``Finite-time distributed consensus in graphs with time-invariant topologies.'' Each agent in Algorithm 2 can computer the consensus value in finite iteration.

Q3. Softmax parameterizations enjoy useful smoothness and boundedness properties (bounded score functions, Lipschitz gradients,
etc.) that make them particularly amenable to rigorous policy-gradient analysis and to
establishing an $\mathcal{O}(1/T)$ rate. However, for deterministic or Gaussian policies, the smooth property is difficult to establish.

We hope that the clarifications provided above adequately address the reviewer’s concerns, and we would be grateful if these revisions could be taken into account in a more favorable reassessment of the manuscript’s rating.

---

> ### Comment · Reviewer_atGW · 2025-11-27
>
> Thank you for the response. After reading the response, most of my concerns are solved, but I still feel doubtful about two parts (W3, Q2). (PS: I think the author's replies are misplaced, the reply under my review belongs to another reviewer.)
>
> W1: After checking proof of Theorem 2 and eq.(21), my concern is addressed. Thank you.
>
> W2: I understand the limitation of stage-wise decomposition technique, which prior work also face. A small suggestion (not necessary): maybe you could further conduct $\gamma>0.95$ to empirically verify the point at which the approximation error degrades performance significantly.
>
> W3: **My concern hasn't been solved.** While the consensus forms a finite-time subroutine, strictly speaking, the total communication complexity is the product of outer iterations and inner consensus steps. Could you provide an explicit expression for the total communication complexity to reach an $\epsilon$-Pareto stationary solution?
>
> Q1: I understand. Thank you.
>
> Q2: **My concern hasn't been solved.** The response addresses the possibility of achieving exact consensus in finite time, but does not answer the specific question: How does the convergence error bound change analytically if we strictly limit the consensus subroutine to a fixed, small number of steps $K_{fixed}$ where exact consensus is NOT achieved?
>
> Q3: I have understood, thank you.
>
> If these two concerns can be fully addressed, I will feel more postive about this paper.

---

> > ### Author Response · Authors · 2025-12-02
> >
> > W3. To address the reviewer’s concern, we provide an explicit expression for the total communication complexity. Let $M\_{\mathrm{finite}}$ denote the number of iterations required by the finite-time consensus subroutine, and let
> > $T(\varepsilon)$ be the number of outer iterations needed to achieve an $\varepsilon$-Pareto stationary solution.
> > Suppose the policy parameter dimension of agent $i$ is $|\theta\_{i}|$. Then, the communication complexity of agent $i$ in the algorithm is $T(\varepsilon)(M\_{\mathrm{finite}}\sum\_{j\in\mathcal{N}\_{i}}|\theta\_{j}|+BH|\mathcal{S}\_{\mathcal{N}^{\kappa}\_{i}}||\mathcal{A}\_{\mathcal{N}^{\kappa}\_{i}}|)$.
> >
> > Q2. In Algorithm~2, agents first need to obtain a consistent estimate of the policy gradient; the subsequent $K_{\lambda}$ inner iterations are then performed to compute an approximate solution of problem (15) based on this consensus estimate. If the consensus step does not reach an exact agreement, the agents effectively solve an approximate version of problem (15). Consequently, the resulting weight $\widehat{\lambda}\_{t}$ carries a bias that is difficult to characterize analytically, because the optimization problem being solved is itself perturbed by the residual consensus error. This bias in $\widehat{\lambda}_{t}$ further propagates into the next policy-gradient update, amplifying the overall gradient
> > approximation error. Therefore, without sufficiently accurate consensus, it becomes challenging to provide a clean analytical bound on the resulting stationarity gap.

---

### Meta-Review · Area_Chair_57f1 · 2026-01-07

**Summary:**

This paper studies multi-objective multi-agent reinforcement learning, by proposing a new distributed actor-critic-type algorithm. The algorithm only uses \kappa-hop neighborhood state information, and then used action-averaged Q-function and linear function approximation, together with a consensus-based procedure to update the multi-objective weights. The convergence rate was then established. It reached a consensus that the localization construction is principled, and the theory is nontrivial. However, there were non-trivial concerns regarding the scalability and the communication realism, as the communication cost and consensus residual were not integrated into the complexity analysis, the limited baselines and incomplete related-work positioning, and the unclear experiment details. The rebuttal did not fully address the major concerns, especially the concern regarding consensus residual. I recommend that the authors incorporate the feedback from this round in preparing the next version of the paper.

**Reviewer Concerns:**

Concerns regarding the Eq. (21)/Theorem 2 on the “bundled” approximation error, the total communication complexity expression, and the incompleteness of related work, were mostly addressed. However, other concerns, especially the major ones mentioned above, remain outstanding.

**Reviewer Scores:**

Reviewer atGW likely will stay with the score 6, and Reviewer e4Vc explicitly said they will maintain score 4. Reviewer LYAQ may increase the score slightly, but not a solid acceptance.

---

### Decision · Program_Chairs · 2026-01-26

Reject